Journal of Data-centric Machine Learning Research (2025)          Submitted 3/25; Revised 6/25; Published 8/25

# TopoBench: A Framework for Benchmarking Topological Deep Learning

**Lev Telyatnikov**[*1], **Guillermo Bernárdez**[*2], **Marco Montagna**[1], **Mustafa Hajij**[3], **Martin Carrasco**[4], **Pavlo Vasylenko**[5], **Mathilde Papillon**[2], **Ghada Zamzmi**[6], **Michael T. Schaub**[7], **Jonas Verhellen**[8], **Pavel Snopov**[9], **Bertran Miquel-Oliver**[10,11], **Manel Gil-Sorribes**[12], **Alexis Molina**[12], **Victor Guallar**[10,13], **Theodore Long**[14], **Julian Suk**[15], **Patryk Rygiel**[15], **Alexander Nikitin**[16], **Giordan Escalona**[17], **Michael Banf**[18], **Dominik Filipiak**[19,18], **Max Schattauer**[18], **Liliya Imasheva**[18], **Alvaro Martinez**[20], **Halley Fritze**[21], **Marissa Masden**[22], **Valentina Sánchez**[23], **Manuel Lecha**[24], **Andrea Cavallo**[25], **Claudio Battiloro**[26], **Matt Piekenbrock**[27], **Mauricio Tec**[26], **George Dasoulas**[26], **Nina Miolane**[2], **Simone Scardapane**[1], **Theodore Papamarkou**[28]

[1]Sapienza University of Rome, [2]UC Santa Barbara, [3]VU Amsterdam, [4]University of Fribourg, [5]Instituto Superior Técnico, [6]University of South Florida, [7]RWTH Aachen University, [8]University of Copenhagen, [9]University of Texas Rio Grande Valley [10]Barcelona Supercomputing Center, [11]Universitat Politècnica de Catalunya, [12]Nostrum Biodiscovery, [13]Catalan Institution for Research and Advanced Studies, [14]Atalaya Capital Management, [15]University of Twente, [16]Aalto University, [17]University of Rochester, [18]Perelyn GmbH, [19]Adam Mickiewicz University, [20]Columbia University, [21]University of Oregon, [22]University of Puget Sound, [23]Tilburg University, [24]Istituto Italiano di Tecnologia, [25]Delft University of Technology, [26]Harvard University, [27]Northeastern University, [28]PolyShape

**Reviewed on OpenReview:** `https://openreview.net/forum?id=07sTzyEVtY`

**Editor:** Yi Liu

## Abstract

This work introduces `TopoBench`, an open-source library designed to standardize benchmarking and accelerate research in topological deep learning (TDL). `TopoBench` decomposes TDL into a sequence of independent modules for data generation, loading, transforming and processing, as well as model training, optimization and evaluation. This modular organization provides flexibility for modifications and facilitates the adaptation and optimization of various TDL pipelines. A key feature of `TopoBench` is its support for transformations and lifting across topological domains. Mapping the topology and features of a graph to higher-order topological domains, such as simplicial and cell complexes, enables richer data representations and more fine-grained analyses. The applicability of `TopoBench` is demonstrated by benchmarking several TDL architectures across diverse tasks and datasets.

**Keywords:** Benchmark, topological deep learning, topological neural networks.

* Equal contribution.

## 1 Introduction

In geometric deep learning (GDL; Bronstein et al., 2021), graph neural networks (GNNs; Zhou et al., 2020) have demonstrated impressive capabilities in processing relational data represented as graphs. However, because graphs represent relationships through edges, they inherently capture only pairwise interactions, which can be a limiting factor. For example, social interactions often involve groups of individuals rather than just pairs, and electrostatic interactions in proteins can span multiple atoms. Topological deep learning (TDL; Papamarkou et al., 2024; Bodnar, 2023; Hajij et al., 2023b; Papillon et al., 2023) offers a framework for modeling complex systems characterized by such multi-way relations among components, leveraging to that end higher-order discrete topological domains (such as simplicial and cell complexes, see Section 2). Topological neural networks (TNNs; Feng et al., 2019; Bunch et al., 2020; Hajij et al., 2020; Bodnar et al., 2021a; Ebli et al., 2020; Schaub et al., 2021; Bodnar et al., 2021b; Chien et al., 2021), which are part of TDL, have found applications in numerous fields that involve higher-order relational data such as social networks (Knoke and Yang, 2019), protein biology (Jha et al., 2022), physics (Wei and Fink, 2024), and computer networks (Bernárdez et al., 2025). TNNs have also shown their potential in various machine learning tasks (Dong et al., 2020; Barbarossa and Sardellitti, 2020; Chen et al., 2022; Roddenberry et al., 2021; Telyatnikov et al., 2025; Giusti et al., 2023).

However, as identified in a recent position paper (Papamarkou et al., 2024), the rapid growth of TDL research has introduced challenges in ensuring reproducibility and conducting systematic comparative evaluations of TNNs. To address these challenges, this work introduces `TopoBench` [1], an open-source and modular framework for TDL. By providing a comprehensive pipeline –from data integration and processing to modeling and evaluation–, our proposed framework facilitates both development and benchmarking of TNNs (Figure 1 illustrates the overall workflow). More specifically, `TopoBench` directly addresses the following relevant limitations of current TDL models' evaluations (Papamarkou et al., 2024):

**Data availability:** Although many complex systems exhibit higher-order interactions, they are mostly collected in the form of point clouds or graphs, implying the failure to fully capture a more nuanced interplay. For instance, in a social network, we might track friendships between individuals but overlook whether they belong to the same group, losing valuable higher-order relationships. This limitation arises because current experimental designs often impose constraints on what data can be collected, making it difficult to systematically capture complex, multi-level relationships. `TopoBench` mitigates the scarcity of higher-order data in three ways. First, it provides an interface for uploading publicly available higher-order datasets. Second, it facilitates the loading of user-defined datasets – whether higher-order or not. Third, it implements lifting algorithms (i.e. mappings between different discrete topological domains) to automate the construction of new topological datasets.

**Standardization:** There is a broad spectrum of TNNs in the TDL literature, each using distinct techniques to preprocess and encode data within a specific higher-order topological domain. This diversity complicates performance comparisons between models on different datasets. To address this issue, `TopoBench` implements a unifying pipeline for data preprocessing and predictive performance evaluation metrics.

---

1. `https://github.com/geometric-intelligence/TopoBench`

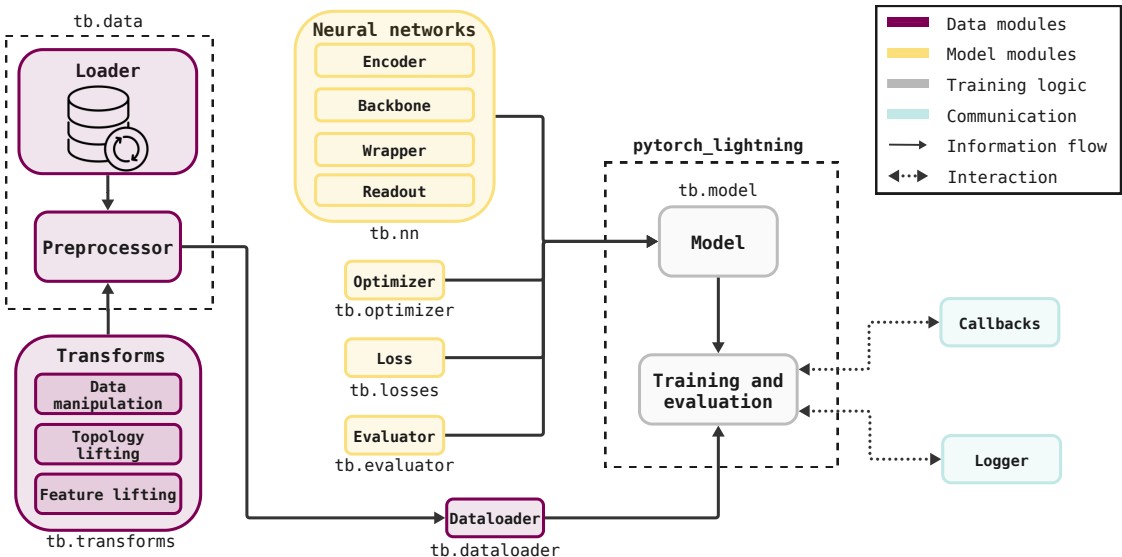

Figure 1: Workflow of `TopoBench`, consisting of four main components: data modules, model modules, training modules, and communication modules.

**Benchmarking:** The described challenges collectively impede the establishment of standardized benchmarking practices within the TDL community. This work provides the first cross-domain benchmarking of TNNs across diverse datasets, adhering to a well-established and rigorous machine learning pipeline. Furthermore, `TopoBench` ensures the complete reproducibility of the experiments.

**Democratization of TDL:** The emerging nature of TDL, coupled with its reliance on advanced mathematical and computer science expertise, poses a barrier to broader adoption. `TopoBench` democratizes TDL by automating and modularizing the pipeline, offering a high-level interface to simplify coding, facilitating seamless integration through a modular design, and ensuring complete compatibility with the `PyTorch` ecosystem. It provides an accessible testbed for newcomers to experiment with topological domains, models, and datasets, fostering innovation and expanding the scope of TDL applications.

The remainder of this paper is structured as follows: Section 2 introduces key TDL concepts –with technical details in the appendix. Section 3 provides a review of related software. Section 4 details `TopoBench`'s modules and functionality. Section 5 demonstrates `TopoBench` through benchmarking experiments. Section 6 concludes with remarks and future directions.

## 2 Background

This section aims to build the general intuition necessary to understand `TopoBench`, while providing references to its formal mathematical foundations.

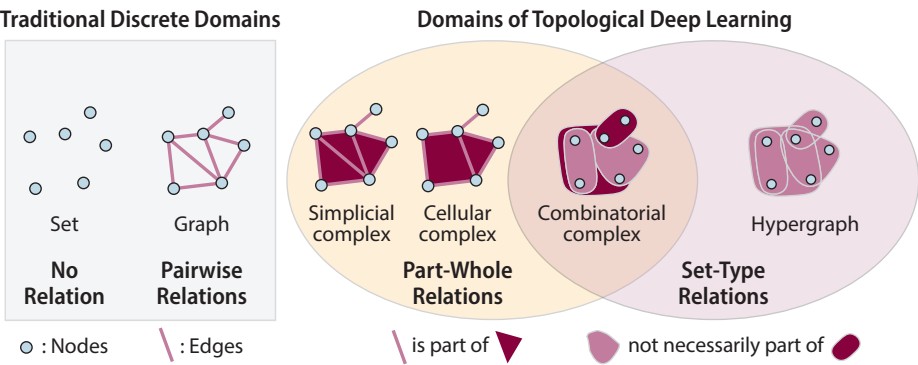

Figure 2: Topological Deep Learning Domains. Nodes in blue, (hyper)edges in pink, and faces in dark red. Figure adapted from Papillon et al. (2023).

**Topological domains.** Relational data can be represented in various forms, with graph representation being the most common framework. However, as discussed in the Introduction, graphs are limited to pairwise relations. TDL methodologies overcome this constraint by encoding higher-order relationships through combinatorial and algebraic topology concepts. Fig. 2 illustrates the standard discrete, higher-order topological spaces used to that end, which enable more complex relational representations via part-whole and set-types relations (Papillon et al., 2023); see Appendix A.1 for the formal definition of each of these topological domains.

**Liftings.** Since most relational data is traditionally collected in discrete domains, such as point clouds and graphs, transitioning to richer topological representations requires mappings between domains — for instance, from a graph to a simplicial complex. This process of mapping, known as lifting, enables more flexible and expressive data representations (further details in Section 4.3 and Appendix A.2).

**Topological neural networks.** Once the data is represented within a chosen topological domain, the TDL pipeline employs neural networks specifically designed for that domain. These models process higher-order structures, leveraging specialized inductive biases. Such networks, referred to as Topological Neural Networks (TNNs), enable learning directly from data represented through topological domains (see Appendix A.3). In general, TNNs exploit a higher-order message-passing mechanism (see Appendix A.5), which generalizes the traditional graph-based message-passing approach (see Appendix A.4), allowing for more comprehensive information propagation through higher-order structures.

## 3 Existing Software

Graph-based learning and GDL are supported by several software packages, including `NetworkX` (Hagberg et al., 2008), `KarateClub` (Rozemberczki et al., 2020), `PyG` (Fey and Lenssen, 2019), `DGL` (Wang et al., 2019), and Open Graph Benchmark (OGB; Hu et al., 2020, 2021). `NetworkX` enables computations on graphs, while `KarateClub` implements unsupervised learning algorithms for graph-structured data. `PyG` and `DGL` provide functionality for GDL as well as standard graph-based learning. Lastly, OGB provides a collection of

graph datasets and a benchmarking framework that supports reproducible graph machine learning research; however, it does not address TDL-specific needs.

Various tools also exist for higher-order domains. For hypergraphs, simplicial complexes, and other topological structures, `HyperNetX` (Liu et al., 2021), `XGI` (Landry et al., 2023), `DHG` (Feng et al., 2019), and `TopoX` (Hajij et al., 2024) each focus on different facets. `HyperNetX` facilitates hypergraph computations, whereas `XGI` supports both hypergraphs and simplicial complexes. `DHG` implements deep learning algorithms for graphs and hypergraphs. `TopoX` is a suite of three packages—`TopoNetX`, `TopoEmbedX`, and `TopoModelX`—that provide broader support for hypergraphs, simplicial, cellular, path, and combinatorial complexes (Hajij et al., 2023a). `TopoNetX` facilitates constructing and computing on these domains, including working with nodes, edges, and higher-order cells; `TopoEmbedX` embeds higher-order domains into Euclidean spaces, while `TopoModelX` implements most TNNs surveyed in Papillon et al. (2023).

Additionally, topological data analysis (TDA) libraries such as `GUDHI` (The GUDHI Project, 2015), `giotto-tda` (Tauzin et al., 2021), and `scikit-tda` (Nathaniel Saul, 2019) offer robust tools for topological computations, like persistent homology diagrams and topological invariant metrics. These TDA packages can provide valuable building blocks to extract topological information from data within TDL pipelines.

## TopoBench Contextualization

`TopoBench` leverages and extends this existing software ecosystem to provide a unified benchmarking infrastructure for TDL. The framework directly integrates established libraries including `NetworkX` for graph computations and the `TopoX` suite—`TopoNetX` for higher-order structure construction and `TopoModelX` for TNN implementations. `TopoBench` also incorporates graph-based models from `PyG` and enables seamless integration of models from original research repositories, providing unprecedented flexibility for TDL evaluation.

While these existing packages provide essential building blocks, `TopoBench` introduces novel capabilities that address critical gaps in the TDL software ecosystem. Unlike OGB's focus on graph learning, `TopoBench` provides comprehensive data management for topological domains, including automated dataset downloading, storage, and processing capabilities. The framework introduces automated lifting transformations that extend beyond `TopoNetX`'s manual construction capabilities, enabling seamless data connectivity transformations between topological domains with integrated feature handling. Additionally, `TopoBench` offers unified mini-batching across all topological structures through a shared dataloader and streamlined configuration systems for experiment setup—capabilities absent from current TDL software.

These innovations collectively establish `TopoBench` as the first comprehensive benchmarking framework for TDL. The framework's unified data representation enables consistent treatment of diverse topological structures, allowing researchers to evaluate models across different domains using standardized procedures. This approach transforms the fragmented TDL software landscape into a cohesive research environment, providing the reproducible benchmarking infrastructure that the rapidly evolving field requires.

---

**Algorithm 1** Execution pipeline for model training in `TopoBench`

---
1:  **Input:** General `cfg` configuration file
2:  `dataset ← Loader(cfg.dataset)`                          # Dataset loading
3:  `splits ← PreProcessor(dataset, cfg.transforms)`     # Transforms and splits
4:  `dataloader ← Dataloader(dataset)`                       # Batch generator
5:  `model ← Model(`                                          # Model initialization
6:      `nn.Encoder(cfg.model),`
7:      `nn.Backbone(cfg.model),`
8:      `nn.BackboneWrapper(cfg.model),`
9:      `nn.Readout(cfg.model),`
10:     `*[Evaluator(cfg.evaluator), Optimizer(cfg.optimizer), Loss(cfg.loss)]`
    `)`
11: `trainer ← lightning.Trainer(cfg.trainer, cfg.callbacks, cfg.logger)`

12: **Model training:**
13: `trainer.fit(model, dataloader)`                         # Model training

14: **Model step for each `batch`:**
15: `batch ← self.encoder(batch)`                            # Feature encoder
16: `model_out ← self.forward(batch)`                        # TNN
17: `model_out ← self.readout(model_out, batch)`             # Readout
18: `model_out ← self.loss(model_out, batch)`                # Loss computation
19: `self.evaluator.update(model_out)`                       # Evaluator update

---

## 4 The TopoBench Library: Module Outline, Datasets and Liftings

`TopoBench` implements a unified and flexible workflow that facilitates the addition of new datasets, data manipulation and preprocessing methods (collectively referred to as transforms), deep learning models, as well as custom metrics and losses. This design ensures applicability across a wide range of tasks and enables a broad cross-domain comparison, currently lacking in the TDL literature. Each module within `TopoBench` is assigned a distinct role while maintaining a consistent input-output structure, which provides a modular interface across all topological domains. Figure 1 outlines the `TopoBench` modules, grouped by functionality into data, model, training, and communication components. Algorithm 1 illustrates the `TopoBench` execution pipeline in pseudo-code.

### 4.1 TopoBench Modules

**Data modules.** These modules manage and process data within `TopoBench`, including `Loader`, `Transforms`, `PreProcessor`, and `Dataloader`.

**Loader.** The `Loader` module provides an interface for downloading and storing data, built upon the widely adopted `InMemoryDataset` from `PyG`, enhancing interoperability. The project webpage offers detailed tutorials on the library, including a step-by-step guide to integrating customized data with these interfaces.

**Transforms.** `Transforms` modules are implemented as subclasses of `BaseTransform` (provided by `PyG`) and include three categories: data manipulation, topology lifting, and feature lifting. The data manipulation module enables general data transformations (e.g., adapting `PyG` (Fey and Lenssen, 2019) or `TopoX` (Hajij et al., 2024) transforms for use in `TopoBench`). The topology lifting and feature lifting modules handle the conversion of data from one topological domain to another (see Section 4.3). Each transform accepts a `Data` object as input, performs the necessary computations, and outputs the modified `Data` object. These composable operators can be easily customized for various tasks.

**Pre-processor.** The `PreProcessor` class applies a sequence of transforms to a dataset. It accepts a dataset object and a list of transforms, iterating over the dataset to apply each transform in turn. To avoid re-computing the same transforms repeatedly, the preprocessed dataset is saved in a dedicated folder for each transform configuration. This setup ensures that each dataset is processed only once per configuration, mitigating the potentially time-consuming nature of preprocessing large datasets. `PreProcessor` also generates or loads data splits according to a chosen strategy (e.g., random splits with predefined proportions, k-fold cross-validation, or fixed splits).

**Dataloader.** The `Dataloader` module provides a consistent interface for batch training across graphs, hypergraphs, simplicial complexes, cell complexes, and combinatorial complexes. By supporting mini-batching for all these domains, it helps make training more tractable on large datasets.

**Model modules.** The neural network modules form the core of the modeling pipeline. The `encoder` component maps initial data features into a latent space and applies a learnable transformation before passing the data to a TNN model –thus standardizing the input across all models. The `backbone` TNN can be imported from existing PyTorch libraries (e.g., `TopoX` or `PyG`), or built on a custom basis within `TopoBench` (see Table 10 in Appendix C). The `wrapper` ensures the correct input is provided to the forward pass of the `backbone` TNN model and collects the output in a dictionary. This design streamlines input and output handling across different topological domains, making it easier to integrate new models into `TopoBench`.

The `readout` module converts latent representations from the neural network into final predictions. The `Loss` module defines a loss function (from the `PyTorch` library, or customized), while the `Optimizer` module configures the optimizer and scheduler. This design allows seamless use of any optimizer and scheduler from `torch.optim`, thereby supporting flexible and robust training. Finally, the `evaluator` module, built upon `torchmetrics`, provides metrics for both classification and regression tasks –while also allowing for tailored ones for specific datasets and tasks. Notably, the flexibility of these modules enable researchers to implement topology-specific evaluation criteria as needed for their particular applications.

**Training and communication modules.** The `Model` class defines a training pipeline for all domains (see lines 14–19 of Algorithm 1). Inheriting from `LightningModule`, it requires `Encoder`, `Wrapper`, `Backbone`, `Readout`, `Evaluator`, `Loss`, and `Optimizer` objects as inputs. The `lightning.Trainer` then automates training, evaluation, and testing. Additional functionalities can be incorporated via callbacks, and users can monitor training with various loggers (e.g., wandb, tensorboard). Both are standard tools in `Lightning` and are referred to as communication modules in `TopoBench`.

## 4.2 Datasets

`TopoBench` includes a wide selection of datasets to accommodate both standard graph-based and higher-order domains. It is the first framework to enable the creation of reliable, reproducible higher-order datasets through the use of various lifting mappings. A subset of these datasets are also used in the experiments of Section 5 for demonstration purposes. See Appendix C.3 for descriptive statistics of the datasets.

**Graph-based datasets.** A number of well-known datasets commonly used in graph-based learning are supported. Citation networks such as Cora, Citeseer, and PubMed (Yang et al., 2016) are included, along with heterophilous datasets (where nodes connected by an edge predominantly belong to different categorical classes), such as Amazon Ratings, Roman Empire, Minesweeper, Tolokers, and Questions. The TU datasets, including MUTAG, PROTEINS, NCI1, NCI109, IMDB-BIN, IMDB-MUL, and REDDIT (Morris et al., 2020), are also integrated, as are molecule datasets like ZINC (Gómez-Bombarelli et al., 2018) and AQSOL (Dwivedi et al., 2023). Furthermore, `TopoBench` supports the US County Demographics dataset (Jia and Benson, 2020), illustrating its adaptability to various graph structures.

**Datasets with higher-order interactions.** Several datasets with higher-order interactions are included in `TopoBench`, showcasing its capabilities to handle data supported on hypergraphs, simplicial complexes, and other topological domains. The MANTRA dataset (Ballester et al., 2024) is part of `TopoBench`, offering over 43,138 two-dimensional and 249,000 three-dimensional triangulations of surfaces and manifolds, which can be used, for example, as features on a simplicial complex. In addition, the widely used AllSet hypergraph datasets (Chien et al., 2021)—Cora-Cocitation, Citeseer-Cocitation, PubMed-Cocitation, Cora-Coauthorship, and DBLP-Coauthorship—are integrated, following the same preprocessing as HyperGCN (Yadati et al., 2019). These hypergraphs group documents co-authored or co-cited together into single hyperedges. Collectively, these examples illustrate how `TopoBench` supports data beyond traditional graph pairwise interactions.

**Compatibility and custom datasets.** To simplify dataset integration, `TopoBench` provides convenient wrappers that build on `PyG` loaders (e.g., `TUDatasets`, `Planetoid`, `ZINC`). In many cases, these wrappers enable straightforward use of any graph dataset already supported by `PyTorch Geometric`, as well as newly introduced datasets such as MANTRA, the hypergraph citation networks, and Human3.6m. Support for custom datasets is facilitated by a simple interface with two key methods: `download()`, for fetching or extracting raw files, and `process()`, for converting the data into the desired relational structure (graph, hypergraph, simplicial or cell complex). Code examples and tutorials provided in `TopoBench` illustrate the `TopoBench` interface for loading custom user-defined datasets[2]. This approach guarantees users can easily extend `TopoBench` to any dataset of interest, thus maintaining the library's modular and extensible design.

## 4.3 Topological Liftings

In the context of TDL, as outlined in Section 2, liftings facilitate the mapping of data from one topological representation to another. This mapping comprises two key aspects:

---

2. `https://github.com/geometric-intelligence/TopoBench/blob/main/tutorials/tutorial_add_ custom_dataset.ipynb`

*structural lifting* and *feature lifting* (see Figure 3 for a visual example, and a formal definition can be found in Appendix A.2). Informally, the *structural lifting* is responsible for the transformation of the underlying relationships or elements of the data. For instance, it might determine how nodes and edges in a graph are mapped into triangles and tetrahedra in a simplicial complex. This structural transformation can be further categorized into *connectivity-based*, where the mapping relies solely on the existing connections within the data, and *feature-based*, where the data's inherent properties or features guide or even fully determine the new structure. Feature lifting, conversely, addresses the transfer of data attributes or features during mapping, ensuring that the properties associated with the data elements are consistently preserved in the new representation, thus maintaining information integrity. Both structural and feature liftings are crucial for the effective application of TDL to diverse and complex datasets.

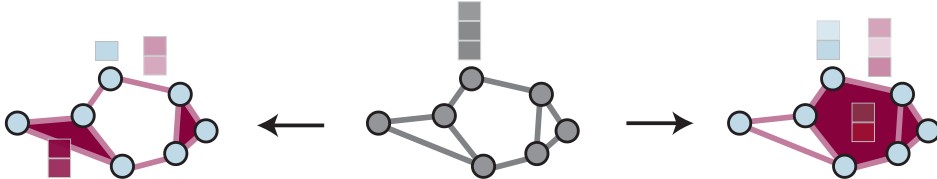

Figure 3: An illustration of lifting a graph (center) to two different topological domains: a simplicial complex (left) and a cell complex (right). The structural lifting maps the nodes and edges of the graph to higher-order topological structures, such as faces, while the feature lifting ensures the associated feature functions are consistently transferred between domains.

Table 3 in Appendix B provides a comprehensive list of all the liftings currently implemented in `TopoBench`. Currently, `TopoBench` supports 11 structural liftings targeting simplicial complexes; 2 targeting cell complexes; 10 moving to hypergraphs; and 3 from a domain to combinatorial complexes. Notably, `TopoBench`'s modular design simplifies the integration of additional liftings, ensuring the framework's adaptability to evolving research needs.

## 5 Numerical Experiments

This section presents numerical experiments that illustrate the breadth of `TopoBench`'s functionality by performing a cross-domain comparison. The overall setup is first described, then results from benchmarking various graph, hypergraph, and TNNs are reported, and an ablation study on signal propagation is presented to demonstrate how `TopoBench` supports comparisons in TDL.

### 5.1 Setup

**Learning tasks and datasets.** Four types of tasks are considered: node classification (seven datasets), node regression (seven datasets), graph classification (seven datasets), and graph regression (one dataset). For node classification, the cocitation datasets (Cora,

Citeseer, and PubMed) and heterophilic datasets (Amazon Ratings, Minesweeper, Roman Empire, and Tolokers) are used (Platonov et al., 2023). For node regression, the election, bachelor, birth, death, income, migration, and unemployment datasets from US election map networks are adapted (Jia and Benson, 2020). In these datasets, each node represents a US state, edges connect neighboring states, and each state is characterized by demographic and election statistics. For each dataset, one statistic is designated as the target, while the others serve as node features, with the dataset named for the chosen target statistic. For graph classification, the TUDataset collection is used, specifically MUTAG, PROTEINS, NCI1, NCI109, IMDB-BIN, IMDB-MUL, and REDDIT (Morris et al., 2020). For graph regression, the ZINC dataset is employed (Irwin et al., 2012).

Higher-order datasets are constructed by lifting these graph datasets. For demonstration purposes, one structural lifting is considered for each of the considered higher-order topological spaces: cycle-based lifting for the cell domain (see Example 5), clique complex lifting for the simplicial domain (see Example 6), and $k$-hop lifting for the hypergraph domain (see Example 8). As for the feature lifting, the projected sum is always considered in all of these scenarios. Descriptive statistics for these topological versions of the datasets are provided in Table 4 in Appendix C.3.

**Models.** Twelve neural networks, supported across four domains (graphs, hypergraphs, simplicial complexes, and cell complexes), are benchmarked. These include three GNNs (GCN, GIN, and GAT), three hypergraph neural networks (EDGNN, AllSetTransformer, and UniGNN2), three simplicial neural networks (SCN, SCCN, and SCCNN), and three cell complex neural networks (CCXN, CWN, and CCCN). Details on these architectures and their hyperparameters appear in Appendix C. In particular, the number of learnable parameters for each best model configuration can be found in Table 8, while the corresponding runtimes are provided in Table 9.

**Training and evaluation.** Five splits are generated for each dataset, with $50\%/25\%/25\%$ of the data going to the training, validation, and test sets, respectively; the exception is ZINC, for which the predefined splits are used (Irwin et al., 2012). The optimal hyperparameter configuration is chosen by selecting the best average performance over the five validation sets (details in Appendix C.2). One performance metric is reported per dataset. Specifically, predictive accuracy is used for Cora, Citeseer, PubMed, Amazon, Roman Empire, MUTAG, PROTEINS, NCI1, NCI109, IMDB-BIN, IMDB-MUL, and REDDIT; AUC-ROC is used for Minesweeper and Tolokers; mean squared error (MSE) is used for election, bachelor, birth, death, income, migration, and unemployment; and mean absolute error (MAE) is used for ZINC. For each dataset, the mean and standard deviation of the chosen metric are computed across the five test sets and reported in Table 1 (where OOM stands for 'out of memory').

## 5.2 Main Results

As seen from Table 1, higher-order neural networks (based on hypergraphs, simplicial, and cell complexes) achieve the best performance on fifteen of twenty-two datasets, whereas GNNs achieve the best performance on six datasets, and tie on the Unemployment dataset. GNNs perform best on node regression in the majority of cases (five out of seven). These best results obtained by GNNs are closely matched by TNNs, since the latter achieve metrics within one standard deviation from the former. In contrast, in nine out of sixteen datasets

Table 1: Cross-domain comparison: results are shown as mean and standard deviation. The best result is bold and shaded in grey, while those within one standard deviation are in blue-shaded boxes.

| | Dataset | GCN | GIN | GAT | AST | EDGNN | UniGNN2 | CWN | CCCN | SCCNN | SCN |
|---|---|---|---|---|---|---|---|---|---|---|---|
| Node-level tasks | Cora | 87.09 ± 0.20 | 87.21 ± 1.89 | 86.71 ± 0.95 | **88.92 ± 0.44** | 87.06 ± 1.09 | 86.97 ± 0.88 | 86.32 ± 1.38 | 87.44 ± 1.28 | 82.19 ± 1.07 | 82.27 ± 1.34 |
| | Citeseer | 75.53 ± 1.27 | 73.73 ± 1.23 | 74.41 ± 1.75 | 73.85 ± 2.21 | 74.93 ± 1.39 | 74.72 ± 1.08 | 75.20 ± 1.82 | **75.63 ± 1.58** | 70.23 ± 2.69 | 71.24 ± 1.68 |
| | Pubmed | 89.40 ± 0.30 | 89.29 ± 0.41 | 89.44 ± 0.24 | **89.62 ± 0.25** | 89.04 ± 0.51 | 89.34 ± 0.45 | 88.64 ± 0.36 | 88.52 ± 0.44 | 88.18 ± 0.32 | 88.72 ± 0.50 |
| | Amazon | 49.56 ± 0.55 | 49.16 ± 1.02 | 50.17 ± 0.59 | 50.50 ± 0.27 | 48.18 ± 0.09 | 49.06 ± 0.08 | **51.90 ± 0.15** | 50.26 ± 0.17 | OOM | OOM |
| | Empire | 78.16 ± 0.32 | 79.56 ± 0.20 | 84.02 ± 0.51 | 79.50 ± 0.13 | 81.01 ± 0.24 | 77.06 ± 0.20 | 81.81 ± 0.62 | 82.14 ± 0.00 | **89.15 ± 0.32** | 88.79 ± 0.46 |
| | Minesweeper | 87.52 ± 0.42 | 87.82 ± 0.34 | 89.64 ± 0.43 | 81.14 ± 0.05 | 84.52 ± 0.05 | 78.02 ± 0.00 | 88.62 ± 0.04 | 89.42 ± 0.00 | 89.0 ± 0.00 | **90.32 ± 0.11** |
| | Tolokers | 83.02 ± 0.71 | 80.72 ± 1.19 | **84.43 ± 1.00** | 83.26 ± 0.10 | 77.53 ± 0.01 | 77.35 ± 0.03 | OOM | OOM | OOM | OOM |
| | Election | 0.31 ± 0.02 | **0.28 ± 0.02** | 0.29 ± 0.02 | 0.29 ± 0.01 | 0.34 ± 0.02 | 0.37 ± 0.02 | 0.34 ± 0.02 | 0.31 ± 0.02 | 0.51 ± 0.03 | 0.46 ± 0.04 |
| | Bachelor | 0.29 ± 0.02 | 0.31 ± 0.03 | **0.28 ± 0.02** | 0.30 ± 0.03 | 0.29 ± 0.02 | 0.31 ± 0.02 | 0.33 ± 0.03 | 0.31 ± 0.02 | 0.34 ± 0.03 | 0.32 ± 0.02 |
| | Birth | 0.72 ± 0.09 | 0.72 ± 0.09 | 0.71 ± 0.09 | 0.71 ± 0.08 | **0.70 ± 0.07** | 0.73 ± 0.10 | 0.72 ± 0.09 | 0.71 ± 0.09 | 0.79 ± 0.12 | 0.71 ± 0.08 |
| | Death | 0.51 ± 0.04 | 0.52 ± 0.04 | 0.51 ± 0.04 | **0.49 ± 0.05** | 0.52 ± 0.05 | 0.51 ± 0.05 | 0.54 ± 0.06 | 0.54 ± 0.06 | 0.55 ± 0.05 | 0.52 ± 0.05 |
| | Income | 0.22 ± 0.03 | 0.21 ± 0.02 | **0.20 ± 0.02** | 0.21 ± 0.02 | 0.23 ± 0.02 | 0.23 ± 0.02 | 0.25 ± 0.03 | 0.23 ± 0.02 | 0.28 ± 0.03 | 0.25 ± 0.02 |
| | Migration | 0.80 ± 0.12 | 0.80 ± 0.10 | **0.77 ± 0.13** | 0.78 ± 0.12 | 0.80 ± 0.12 | 0.79 ± 0.12 | 0.84 ± 0.13 | 0.84 ± 0.12 | 0.90 ± 0.14 | 0.92 ± 0.20 |
| | Unempl | 0.25 ± 0.03 | **0.22 ± 0.02** | 0.23 ± 0.03 | **0.22 ± 0.02** | 0.26 ± 0.03 | 0.28 ± 0.02 | 0.25 ± 0.03 | 0.25 ± 0.03 | 0.43 ± 0.04 | 0.38 ± 0.04 |
| Graph-level tasks | MUTAG | 69.79 ± 6.80 | 79.57 ± 6.13 | 72.77 ± 2.77 | 71.06 ± 6.49 | 80.00 ± 4.90 | 80.43 ± 4.09 | **80.43 ± 1.78** | 77.02 ± 9.32 | 76.17 ± 6.63 | 73.62 ± 6.13 |
| | PROTEINS | 75.70 ± 2.14 | 75.20 ± 3.30 | 76.34 ± 1.66 | **76.63 ± 1.74** | 73.91 ± 4.39 | 75.20 ± 2.96 | 76.13 ± 2.70 | 73.33 ± 2.30 | 74.19 ± 2.86 | 75.27 ± 2.14 |
| | NCI1 | 72.86 ± 0.69 | 74.26 ± 0.96 | 75.00 ± 0.99 | 75.18 ± 1.24 | 73.97 ± 0.82 | 73.02 ± 0.92 | 73.93 ± 1.87 | **76.67 ± 1.48** | 76.60 ± 1.75 | 74.49 ± 1.03 |
| | NCI109 | 72.20 ± 1.22 | 74.42 ± 0.70 | 73.80 ± 0.73 | 73.75 ± 1.09 | 74.93 ± 2.50 | 70.76 ± 1.11 | 73.80 ± 2.06 | 75.35 ± 1.50 | **77.12 ± 1.07** | 75.70 ± 1.04 |
| | IMDB-BIN | **72.00 ± 2.48** | 70.96 ± 1.93 | 69.76 ± 2.65 | 70.32 ± 3.27 | 69.12 ± 2.92 | 71.04 ± 1.31 | 70.40 ± 2.02 | 69.12 ± 2.82 | 70.88 ± 2.25 | 70.80 ± 2.38 |
| | IMDB-MUL | 49.97 ± 2.16 | 47.68 ± 4.21 | 50.13 ± 3.87 | **50.51 ± 2.92** | 49.17 ± 4.35 | 49.76 ± 3.55 | 49.71 ± 2.83 | 47.79 ± 3.45 | 48.75 ± 3.98 | 49.49 ± 5.08 |
| | REDDIT | 76.24 ± 0.54 | 81.96 ± 1.36 | 75.68 ± 1.00 | 74.84 ± 2.68 | 83.24 ± 1.45 | 75.56 ± 3.19 | **85.52 ± 1.38** | 85.12 ± 1.29 | 77.24 ± 1.87 | 71.28 ± 2.06 |
| | ZINC | 0.62 ± 0.01 | 0.57 ± 0.04 | 0.61 ± 0.01 | 0.59 ± 0.02 | 0.51 ± 0.01 | 0.60 ± 0.01 | **0.34 ± 0.01** | **0.34 ± 0.02** | 0.36 ± 0.02 | 0.53 ± 0.04 |

TNNs outperform GNNs, and attain performance metrics that are higher by more than one standard deviation with respect to GNNs. In other words, in situations where higher-order networks outperform GNNs, the performance gap is more pronounced. It is also noted that, for demonstration purposes, only one fixed lifting is considered to transform graph data to each of the considered topological domains (see Appendix C.4). These results suggest that, even without lifting optimization, TNNs have an advantage over GNNs in terms of performance, although it is worth emphasizing that overall they also tend to be less efficient in terms of memory usage and computational time than graph-based counterparts (see Appendix C.4 for a more detailed analysis). However, and more importantly for the context of this paper, the benchmarks demonstrate the degree of comparisons that can be performed with `TopoBench` across models and datasets.

**Remark.** Notably, OOM results are originated when lifting large, densely connected graphs to higher-order domains, showcasing the scalability issues of the liftings leveraged in this analysis (i.e., clique and cycle liftings to simplicial and cellular domains, respectively).

## 5.3 Ablation Study

This ablation study examines how different readout strategies influence performance in neural networks built on higher-order domains, highlighting the importance of node-level signal updates and pooling choices. First, graph and hypergraph (neural network) models differ from simplicial and cell complex (neural network) models in terms of the domains and, subsequently, representations they support. Graph and hypergraph models can output two types of representations: node representations and edge or hyperedge representations. In contrast, the output of simplicial and cell models depends on the different types of cells present (0-cell up to $n$-cells) and on the model itself. For example, a simplicial or cell complex model may process an $n$-cell input but may not produce an $n$-cell output. The `backbone_wrapper` in `TopoBench` addresses these differences in the underlying domains of the models.

There is a second difference, which is inherent in the TNNs themselves. Consider a downstream classification task. For graphs, the standard practice is to perform classification over pooled node features. However, this aspect has not been extensively studied in the TDL literature. For instance, a simplicial or cell model may update 1-cell representations (edges) or 2-cell representations (cycles or triangles) while leaving 0-cell (node) representations unchanged, making direct pooling over nodes potentially ineffective. One could consider more elaborate update processes in which different $n$-cell representations are combined, but this renders pooling more intricate for higher-order domains. These architectural considerations are complex and remain open research questions in TDL.

Nevertheless, to fairly compare different neural network architectures, this second difference must be addressed. To that end, this ablation study considers two types of readouts to enable a rigorous evaluation: direct readout (DR), where the downstream task is performed directly over the 0-cell representation, and signal down-propagation (SDP), where information from higher-order cell representations is iteratively fused down to 0-cell representations using appropriate incidence matrices, followed by a linear projection over the concatenated $(n-1)$-cell signal and the fused $(n-1)$-cell representation. For instance, if a simplicial or cell complex model outputs 0-cell, 1-cell, and 2-cell representations, the signal propagates from 2-cells to 1-cells and then from 1-cells to 0-cells during readout. The downstream task is then performed over the updated 0-cell representations.[3]

Table 2 shows that the best-performing readout type depends on how a model propagates signals internally. For example, the CWN model does not update 0-cell representations, so the SDP strategy performs notably better. Conversely, CCCN, SCCNN, and SCN propagate information to 0-cells, making SDP readout yield only small or negligible changes in performance. Further details are available in Appendix C.4.

These results underscore the impact of structural properties in TNNs. The performance variations observed in this ablation study emphasize the critical role of architectural and lifting decisions for higher-order learning models. By enabling comparisons across a wide range of models and datasets, `TopoBench` facilitates deeper insights and drives advancements in TDL.

## 5.4 Higher-Order Datasets

Appendix D presents additional illustrative experiments conducted on 13 datasets included in `TopoBench`, spanning a broad range of hypergraph datasets (for classification tasks) and simplicial datasets (for both classification and regression tasks). The evaluation protocol follows the setup described in Section 5.1, with the exception of structural and feature liftings, as these datasets natively possess higher-order topologies and include features on higher-order cells.

For the hypergraph datasets, no single model consistently outperforms others across all benchmarks. Among the evaluated models, AllSetTransformer achieves the best performance on 5 out of 10 datasets. For the simplicial MANTRA family of datasets, the results demonstrate that topological tasks are more effectively modeled by TNNs, whereas standard GNN baselines fail to capture the intricate topological structures, resulting in lower performance on purely topological tasks.

---

3. See Appendix A.3 for an introduction to TNNs and Higher-Order Message Passing on topological domains.

Table 2: This ablation study compares the performance of CWN, CCCN, SCCNN, and SCN models on various datasets using two readout strategies, direct readout (DR) and signal down-propagation (SDP). SDP generally enhances CWN performance, whereas the effect of SDP on CCCN, SCCNN, and SCN varies based on their internal signal propagation mechanisms. Means and standard deviations of performance metrics are shown. The best results are shown in bold for each model and readout type.

| Dataset | CWN | | CCCN | | SCCNN | | SCN | |
|---|---|---|---|---|---|---|---|---|
| | DR | SDP | DR | SDP | DR | SDP | DR | SDP |
| **Node-level tasks** | | | | | | | | |
| Cora | 74.95 ± 0.98 | **86.32 ± 1.38** | 87.44 ± 1.28 | **87.68 ± 1.17** | **82.19 ± 1.07** | 80.65 ± 2.39 | **82.27 ± 1.34** | 79.91 ± 1.18 |
| Citeseer | 70.49 ± 2.85 | **75.20 ± 1.82** | **75.63 ± 1.58** | 74.91 ± 1.25 | **70.23 ± 2.69** | 69.03 ± 2.01 | **71.24 ± 1.68** | 70.40 ± 1.53 |
| Pubmed | 86.94 ± 0.68 | **88.64 ± 0.36** | 88.52 ± 0.44 | **88.67 ± 0.39** | **88.18 ± 0.32** | 87.78 ± 0.58 | **88.72 ± 0.50** | 88.62 ± 0.44 |
| Amazon | 45.58 ± 0.25 | **51.90 ± 0.15** | **50.55 ± 0.15** | 50.26 ± 0.17 | OOM | OOM | OOM | OOM |
| Empire | 66.13 ± 0.03 | **81.81 ± 0.62** | 82.14 ± 0.00 | **82.51 ± 0.00** | **89.15 ± 0.32** | 88.73 ± 0.12 | 85.89 ± 0.34 | **88.79 ± 0.46** |
| Minesweeper | 48.82 ± 0.00 | **88.62 ± 0.04** | 89.42 ± 0.00 | **89.85 ± 0.00** | 87.40 ± 0.00 | **89.00 ± 0.00** | **90.32 ± 0.11** | 90.27 ± 0.36 |
| Election | 0.60 ± 0.04 | **0.34 ± 0.02** | **0.31 ± 0.02** | **0.31 ± 0.01** | **0.51 ± 0.03** | 0.56 ± 0.04 | **0.46 ± 0.04** | 0.51 ± 0.03 |
| Bachelor | **0.33 ± 0.03** | **0.33 ± 0.03** | 0.32 ± 0.02 | **0.31 ± 0.02** | 0.34 ± 0.03 | **0.34 ± 0.03** | **0.32 ± 0.02** | **0.32 ± 0.03** |
| Birth | 0.81 ± 0.11 | **0.72 ± 0.09** | **0.71 ± 0.09** | 0.72 ± 0.05 | 0.79 ± 0.12 | 0.83 ± 0.12 | **0.71 ± 0.08** | 0.80 ± 0.11 |
| Death | 0.55 ± 0.05 | **0.54 ± 0.06** | **0.54 ± 0.06** | **0.54 ± 0.06** | **0.55 ± 0.05** | 0.58 ± 0.05 | **0.52 ± 0.05** | 0.56 ± 0.05 |
| Income | 0.36 ± 0.04 | **0.25 ± 0.03** | **0.23 ± 0.02** | **0.23 ± 0.02** | **0.28 ± 0.03** | 0.31 ± 0.03 | **0.25 ± 0.02** | 0.27 ± 0.02 |
| Migration | 0.90 ± 0.16 | **0.84 ± 0.13** | **0.84 ± 0.10** | 0.84 ± 0.12 | **0.90 ± 0.14** | 0.93 ± 0.17 | **0.92 ± 0.20** | 0.96 ± 0.23 |
| Unempl | 0.46 ± 0.04 | **0.25 ± 0.03** | **0.24 ± 0.03** | 0.25 ± 0.03 | **0.43 ± 0.04** | 0.45 ± 0.04 | **0.38 ± 0.04** | 0.41 ± 0.03 |
| **Graph-level tasks** | | | | | | | | |
| MUTAG | 69.68 ± 8.58 | **80.43 ± 1.78** | **80.85 ± 5.42** | 77.02 ± 9.32 | **76.17 ± 6.63** | 70.64 ± 3.16 | 71.49 ± 2.43 | **73.62 ± 6.13** |
| PROTEINS | **76.13 ± 1.80** | **76.13 ± 2.70** | **73.55 ± 3.43** | 73.33 ± 2.30 | 74.19 ± 2.86 | **74.98 ± 1.92** | **75.27 ± 2.14** | 74.77 ± 1.69 |
| NCI1 | 68.52 ± 0.51 | **73.93 ± 1.87** | 76.67 ± 1.48 | **77.65 ± 1.28** | **76.60 ± 1.75** | 75.60 ± 2.45 | **75.27 ± 1.57** | 74.49 ± 1.03 |
| NCI109 | 68.19 ± 0.65 | **73.80 ± 2.06** | **75.35 ± 1.50** | 74.83 ± 1.18 | **77.12 ± 1.07** | 75.43 ± 1.94 | 74.58 ± 1.29 | **75.70 ± 1.04** |
| IMDB-BIN | **70.40 ± 2.02** | 69.28 ± 2.57 | 69.12 ± 2.82 | **69.44 ± 2.46** | **70.88 ± 2.25** | 69.28 ± 5.69 | **70.80 ± 2.38** | 68.64 ± 3.90 |
| IMDB-MUL | 49.71 ± 2.83 | **49.87 ± 2.33** | **49.01 ± 2.63** | 47.79 ± 3.45 | **48.75 ± 3.98** | 46.67 ± 3.13 | 48.16 ± 2.89 | **49.49 ± 5.08** |
| REDDIT | 76.20 ± 0.86 | **85.52 ± 1.38** | **85.12 ± 1.29** | 83.32 ± 0.73 | 75.56 ± 3.46 | **77.24 ± 1.87** | **71.28 ± 2.06** | 69.68 ± 4.00 |
| ZINC | 0.70 ± 0.00 | **0.34 ± 0.01** | 0.35 ± 0.02 | **0.34 ± 0.02** | **0.36 ± 0.01** | 0.36 ± 0.02 | 0.59 ± 0.01 | **0.53 ± 0.04** |

## 6 Concluding Remarks, Limitations, and Future Work

This paper has introduced `TopoBench`, an open-source benchmarking framework for TDL. By organizing the TDL pipeline into a sequence of modular steps, `TopoBench` simplifies the benchmarking process and accelerates research. A key feature of `TopoBench` is its ability to map graph topology and features to higher-order topological domains such as simplicial and cell complexes, enabling richer data representations and more detailed analyses. In addition, `TopoBench` provides direct access to a wide variety of real and synthetic datasets, covering both graph-based and higher-order domains. The effectiveness of `TopoBench` has been demonstrated by benchmarking several TDL architectures across diverse learning tasks and datasets, offering insights into the relative advantages of different models.

While `TopoBench` already addresses several challenges in TDL, it also has limitations that point to promising directions for future enhancements. One area is the implementation of learnable liftings, which are supported by `TopoBench`. This direction could enable task-specific topological representations learned dynamically from data. A second limitation lies in the broader scarcity of standardized, real-world higher-order datasets. Although `TopoBench` incorporates numerous datasets for hypergraph, simplicial, and cell complexes, this remains an active area of expansion. Providing more built-in higher-order datasets will further streamline research in TDL.

Another potential direction is to perform an exhaustive exploration of optimal liftings per combination of domains, datasets, and models. In fact, OOM values showcase the scalability

limitations of the most common used strategies to lift graphs into simplicial and cellular domains (i.e., clique and cycle liftings, respectively). Finally, extending the set of evaluation metrics beyond classification or regression accuracy to include more TDL-specific measures of expressivity, explainability, and fairness (Papamarkou et al., 2024) is another avenue of growth –and these modules have been designed to be easily extendable.

Moving forward, the modular design of `TopoBench` invites contributions from the community. Researchers and practitioners are encouraged to contribute to `TopoBench` by introducing new learnable liftings, adding datasets, and developing specialized performance metrics. Moreover, to mitigate the aforementioned scalability issues, several strategies can be explored –e.g. pruning the input graphs prior to lifting, employing scalable lifting mechanisms—such as those explored in the ICML 2024 TDL Challenge (Bernárdez et al., 2024)—and applying mini-batching techniques to higher-order structures in transductive settings (analogous to those used in GNN modeling). These efforts will not only strengthen the benchmarking ecosystem of TDL but also help drive innovation in topological deep learning more broadly –as already shown in the recents works of TopoTune (Papillon et al., 2025) and HOPSE (Carrasco et al., 2025), both of which leverage `TopoBench` framework to push the boundaries of TDL.

## Code Availability and Reproducibility

The code for `TopoBench` is publicly available on GitHub under the MIT license: `https://github.com/geometric-intelligence/TopoBench`. The codebase employs continuous integration, is fully documented, and provides comprehensive API documentation at `https://geometric-intelligence.github.io/topobench/index.html`.

All aspects of library installation and development are described in the `README.md` file. To replicate the experiments reported in this paper, refer to the 'Experiments Reproducibility' section in `README.md`. Additional tutorials in the 'Tutorials' section illustrate how to integrate new models, datasets, learnable liftings, and transforms within `TopoBench`.

## Broader Impact Statement

`TopoBench` aims to standardize benchmarking in TDL, thus benefiting the community by facilitating and accelerating research developments in TDL and its applications. We do not expect `TopoBench` to have any direct negative societal impact from its usage. Moreover, the code of conduct for `TopoBench` contributors, which is publicly available in the 'README.md' file of the GitHub repository of the library, sets concrete ethical standards, promotes transparency, fairness, and inclusivity in research.

The `TopoBench` library will be constantly maintained to respect proprietary content. It will implement strict revision processes to ensure that all code implementations, libraries, and datasets have open-source licenses that guarantee their legitimate usage within the framework.

## Author Contributions

L. Telyatnikov and G. Bernárdez contributed equally to this work as the main authors and lead developers. The conceptualization of the TopoBench project was a collaborative effort

by L. Telyatnikov, G. Bernárdez, M. Montagna, N. Miolane, T. Papamarkou, M. Hajij, G. Zamzmi, M. T. Schaub, and S. Scardapane. The core development and implementation of the benchmark were carried out by L. Telyatnikov, G. Bernárdez, M. Montagna, M. Carrasco, P. Vasylenko, M. Papillon, and N. Miolane. The experiments were led by L. Telyatnikov with support from G. Bernárdez. The manuscript was written by T. Papamarkou, L. Telyatnikov, and G. Bernárdez, with significant writing contributions to various sections from S. Scardapane, M. Hajij, G. Zamzmi, and M. T. Schaub. All other authors contributed to the TopoBench ecosystem through their winning submissions (i.e. lifting implementations) to the ICML TDL Challenge 2024 (Bernárdez et al., 2024).

## Acknowledgments and Disclosure of Funding

M. Papillon, G. Bernárdez and N. Miolane acknowledge support from the National Science Foundation, Award DMS-2134241. M. Papillon and N. Miolane acknowledge funding from the National Science Foundation, Award DMS-2240158 and from the Noyce Foundation. M. Papillon acknowledges the support of the Natural Sciences and Engineering Research Council of Canada. M. Hajij acknowledges support from the National Science Foundation, award DMS-2134231. M. T. Schaub acknowledges funding by the European Union (ERC, HIGH-HOPeS, 101039827). Views and opinions expressed are however those of the author(s) only and do not necessarily reflect those of the European Union or the European Research Council Executive Agency. Neither the European Union nor the granting authority can be held responsible for them.

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

## Appendix A. Mathematical Background

Relational data modeling is a fundamental aspect of modern machine learning and data analysis, particularly in domains where complex relationships between entities play a crucial role. This appendix provides a comprehensive overview of the key concepts and techniques in relational data modeling, with a focus on topological approaches that capture intricate structural information. It also provides the essential mathematical background required to effectively use `TopoBench`.

We begin by exploring various topological domains, from the familiar terrain of graphs to more sophisticated structures such as hypergraphs, simplicial complexes, cell complexes, and combinatorial complexes (see Appendix A.1). These domains offer powerful frameworks for representing and analyzing complex relational data[4]. Next, we introduce the *lifting*

---

4. `TopoBench` supports simplicial complexes, cell complexes, hypergraphs, and combinatorial complexes. The `TopoBench` modularity allows for easy addition of other topological domains.

*mechanism*, which enables the mapping of one topological domain onto another, facilitating flexible data representations (refer to Appendix A.2). Finally, we conclude by presenting a mathematical introduction to Topological Neural Networks, which are used to model data represented with the help of one of the topological domains (see Appendix A.3).

## A.1 Topological Domains

This section introduces the topological domains implemented in `TopoBench`, which provide powerful frameworks for modeling complex relationships and structures in data. We begin with the fundamental concept of graphs, laying the groundwork for understanding more intricate structures. From there, we explore higher-order domains — including hypergraphs, simplicial complexes, cell complexes, and combinatorial complexes — each offering unique capabilities for capturing different types of relationships and hierarchies within data.

**Definition 1** *Let $\mathcal{G} = (V, E)$ be a graph, with node set $V$ and edge set $E$. A featured graph is a tuple $\mathcal{G}_F = (V, E, F_V, F_E)$, where $F_V : V \to \mathbb{R}^{d_v}$ is a function that maps each node to a feature vector in $\mathbb{R}^{d_v}$ and $F_E : E \to \mathbb{R}^{d_e}$ is a function that maps each edge to a feature vector in $\mathbb{R}^{d_e}$.*

A topological domain is a generalization of a graph that captures both pairwise and higher-order relationships between entities (Bick et al., 2023; Battiston et al., 2021). When working with topological domains, two key properties come into play: set-type relations and hierarchical structures represented by rank functions (Hajij et al., 2023b; Papillon et al., 2023).

**Definition 2 (Set-type relation)** *A relation in a topological domain is called a set-type relation if its existence is not implied by another relation in the domain.*

**Definition 3 (Rank function)** *A rank function on a higher-order domain $\mathcal{X}$ is an order-preserving function $rk \colon \mathcal{X} \to \mathbb{Z}_{\geq 0}$ such that $x \subseteq y$ implies $rk(x) \leq rk(y)$ for all $x, y \in \mathcal{X}$.*

Set-type relations emphasize the independence of connections within a domain, allowing for flexible representation of complex interactions. In contrast, rank functions introduce a hierarchical (also referred to as part-whole) organization that facilitates the representation and analysis of nested relationships.

**Hypergraphs** Hypergraphs generalize traditional graphs by allowing edges, known as hyperedges, to connect any number of nodes. This flexibility enables hypergraphs to capture more complex relationships between entities than standard graphs, which only connect pairs of nodes. Hypergraphs exhibit set-type relationships that lack an explicit notion of hierarchy. Using these set-type relations makes them a powerful tool for representing relationships across a diverse range of complex systems.

**Definition 4 (Hypergraph)** *A hypergraph $\mathcal{H}$ on a nonempty set $\mathcal{V}$ is a pair $(\mathcal{V}, \mathcal{E}_{\mathcal{H}})$, where $\mathcal{E}_{\mathcal{H}}$ is a non-empty subset of the powerset $\mathcal{P}(\mathcal{V}) \setminus \{\emptyset\}$. Elements of $\mathcal{E}_{\mathcal{H}}$ are called hyperedges.*

**Example 1 (Collaborative Authorship Networks)** *In collaborative networks, authors are represented as nodes, and co-authorship on a paper forms a hyperedge connecting all authors involved.*

**Simplicial complexes**    Simplicial complexes extend graphs by incorporating hierarchical part-whole relationships through the multi-scale construction of cells. In this structure, nodes correspond to rank 0-cells, which can be combined to form edges (rank 1-cells). Edges can then be grouped to form faces (rank 2 cells), and faces can be combined to create volumes (rank 3-cells), continuing in this manner. Consequently, the faces of a simplicial complex are triangles, volumes are tetrahedrons, and higher-dimensional cells follow the same pattern. A key feature of simplicial complexes is their strict hierarchical structure, where each $k$-dimensional simplex is composed of $(k-1)$-dimensional simplices, reinforcing a strong sense of hierarchy across all levels.

**Definition 5 (Simplicial Complex)** *A simplicial complex (SC) in a non-empty set $S$ is a pair $SC = (S, \mathcal{X})$, where $\mathcal{X} \subset \mathcal{P}(S) \setminus \{\emptyset\}$ satisfies: if $x \in SC$ and $y \subseteq x$, then $y \in SC$. The elements of $\mathcal{X}$ are called simplices.*

**Example 2 (3D Surface Meshes)** *3D models of objects, such as those used in computer graphics or for representing anatomical structures, are often constructed using triangular meshes. These meshes naturally form simplicial complexes, where the vertices of the triangles are 0-simplices, the edges are 1-simplices, and the triangular faces themselves are 2-simplices.*

**Cell complexes**    Cell complexes provide a hierarchical interior-to-boundary structure, offering clear topological and geometric interpretations, but they are not based on set-type relations. Unlike simplicial complexes, cell complexes are not limited to simplexes; faces can involve more than three nodes, allowing for a more flexible representation. This increased flexibility grants cell complexes greater expressivity compared to simplicial complexes Bodnar et al. (2021a); Bodnar (2023).

**Definition 6 (Cell complex)** *A regular cell complex is a topological space $S$ partitioned into subspaces (cells) $\{x_\alpha\}_{\alpha \in P_S}$, where $P_S$ is an index set, satisfying:*

1. *$S = \cup_{\alpha \in P_S} int(x_\alpha)$, where $int(x)$ denotes the interior of cell $x$.*

2. *For each $\alpha \in P_S$, there exists a homeomorphism $\psi_\alpha$ (attaching map) from $x_\alpha$ to $\mathbb{R}^{n_\alpha}$ for some $n_\alpha \in \mathbb{N}$. The integer $n_\alpha$ is the dimension of cell $x_\alpha$.*

3. *For each cell $x_\alpha$, the boundary $\partial x_\alpha$ is a union of finitely many cells of strictly lower dimension.*

**Example 3 (Molecular structures.)** *Molecules admit natural representations as cell complexes by considering atoms as nodes (i.e., cells of rank zero), bonds as edges (i.e., cells of rank one), and rings as faces (i.e., cells of rank two).*

**Combinatorial complexes**    Combinatorial complexes combine hierarchical structure with set-type relations, enabling a flexible yet comprehensive representation of higher-order networks.

**Definition 7 (Combinatorial complex)** *A combinatorial complex (CC) is a triple $(\mathcal{V}, \mathcal{X}, \mathrm{rk})$ consisting of a set $\mathcal{V}$, a subset $\mathcal{X} \subset \mathcal{P}(\mathcal{V}) \setminus \{\emptyset\}$, and a function $\mathrm{rk} \colon \mathcal{X} \to \mathbb{Z}_{\geq 0}$ satisfying:*

1. For all $v \in \mathcal{V}$, $\{v\} \in \mathcal{X}$ and $\mathrm{rk}(v) = 0$.

2. The function rk *is order-preserving: if* $x, y \in \mathcal{X}$ *with* $x \subseteq y$*, then* $\mathrm{rk}(x) \leq \mathrm{rk}(y)$*.*

**Example 4 (Geospatial structures.)** *Geospatial data, comprised of grid points (0-cells), road polylines (1-cells), and census tract polygons (2-cells), can be effectively represented using combinatorial complexes. A visual example is provided in Figure 2 (Right) of Battiloro et al. (2025).*

**Featured topological domains.** A featured graph is a graph whose nodes or edges are equipped with feature functions (Sanchez-Lengeling et al., 2021). `TopoBench` generalizes this idea to featured topological domains, where each topological element (e.g., simplex or cell) can carry feature vectors. Although the following definitions use cell complexes as a template, the same ideas apply to other domains (simplicial complexes, hypergraphs, and so on).

**Definition 8 (Featured topological domain)** *A* featured topological domain *is a pair* $(\mathcal{X}, F)$*, where* $\mathcal{X}$ *is a topological domain and* $F = \{F_i\}_{i \geq 0}$ *is a collection of feature functions. Each function* $F_i$ *maps the* $i$*-dimensional elements of* $\mathcal{X}$*, denoted* $\mathcal{X}_i$*, to a feature space* $\mathbb{R}^{k_i}$*:*

$$F_i \colon \mathcal{X}_i \to \mathbb{R}^{k_i}.$$

## A.2 Liftings

Lifting describes the process of mapping two topological domains through a well-defined procedure (Hajij et al., 2023b; Papillon et al., 2023). This work extends this concept by providing a unified mathematical framework that generalizes all lifting procedures from the 2nd Topological Deep Learning Challenge at ICML 2024 (Bernárdez et al., 2024).

**Definition 9 (Lifting between featured topological domains)** *Let* $T_1 = (\mathcal{X}_1, F_1)$ *and* $T_2 = (\mathcal{X}_2, F_2)$ *be two featured topological domains. A* lifting *from* $T_1$ *to* $T_2$ *is a pair* $(\psi_X, \psi_F)$*, where:*

1. **Structural lifting** $\psi_X \colon \mathcal{X}_1 \times F_1 \to \mathcal{X}_2$ *is a map that determines how elements of* $\mathcal{X}_1$ *are mapped into* $\mathcal{X}_2$*.*

2. **Feature lifting** $\psi_F \colon \mathcal{X}_1 \times F_1 \to F_2$ *is a map that transforms feature functions while maintaining consistency with* $\psi_X$*, meaning that for all* $x \in \mathcal{X}_1$*,*

$$F_2(\psi_X(x)) = \psi_F(F_1(x)).$$

In practice, structural liftings can be taxonimized as *connectivity-* and/or *feature-based.* Connectivity-based structural lifting $\psi_X$ maps the elements of $\mathcal{X}_1$ to $\mathcal{X}_2$ relying solely on the given topology $\mathcal{X}_1$. In contrast, feature-based structural lifting leverages the features $F_1$ either to conditionally guide the mapping of topology or to fully infer the topology $\mathcal{X}_2$ from $F_1$. The feature lifting $\psi_F$ further ensures that the associated features are consistently transferred. Examples appear in Figure 3.

### A.2.1 Lifting Examples

In this section, we present four examples of lifting from the graph domain to higher-order topological domains (see Examples 5, 6, 7, and 8), followed by two application examples demonstrating how topological domains can be used to describe real-world data (see Examples 9 and 10).

**Example 5** *From graphs to cell complexes: cycle-based liftings. A graph is lifted to a cell complex in two steps. First, a finite set of cycles (closed loops) within the graph is identified. Second, each identified cycle is associated with a 2-cell whose boundary is exactly that cycle. The nodes and edges of the cell complex are inherited from the original graph.*

**Example 6** *From graphs to simplicial complexes: clique complexes. By lifting a graph to a simplicial complex, both pairwise and higher-order interactions can be captured. For a given graph, the corresponding clique complex is formed by treating every complete subgraph (clique) as a simplex. Specifically, each node is a 0-simplex, each edge (clique of size 2) is a 1-simplex, each triangle (clique of size 3) is a 2-simplex, and so forth. In general, a clique of size $k + 1$ becomes a $k$-simplex.*

**Example 7** *From graphs to simplicial complexes: neighbor complexes. Neighbor complexes lift the neighborhoods of nodes to simplices as follows. For each node in the graph, the node itself and all its neighbors are considered as a single set. This set is then treated as a simplex, whose dimension depends on the node's degree. For instance, if a node has d neighbors, it forms a d-simplex.*

**Example 8** *From graphs to hypergraphs: k-hop liftings. Let $\mathcal{G} = (V, E)$ be a graph and $\mathcal{H} = (V, \mathcal{E})$ be a hypergraph. The k-neighborhood $N_k(v)$ of a node $v \in V$ in $\mathcal{G}$ consists of all nodes reachable within k steps from v. To lift $\mathcal{G}$ to $\mathcal{H}$, a hyperedge $e_v$ is assigned to each node $v \in V$ in $\mathcal{H}$, where $e_v = N_k(v)$. Thus, the set of hyperedges in $\mathcal{H}$ is given by $\mathcal{E} = \{ N_k(v) \mid v \in V \}$.*

**Example 9** *Lifting a Social Network to a Higher-Order Topological Domain. Let $T_1$ be a social network represented as a graph, where nodes correspond to individuals and edges indicate social interactions (e.g., friendships, collaborations, or message exchanges). We lift this structure to a hypergraph or simplicial complex $T_2$, where higher-order interactions capture group dynamics beyond pairwise relationships.*

- *The structural lifting $\psi_X$ maps tightly connected communities or recurring social interactions in $T_1$ to higher-order simplices in $T_2$. For instance, a group of researchers collaborating on multiple papers could be lifted from a clique in $T_1$ to a 3-simplex in $T_2$, representing a collective research effort.*

- *The feature lifting $\psi_F$ aggregates individual attributes (e.g., influence score, topic preferences, engagement level) into group-level properties (e.g., collective expertise, community sentiment, or information diffusion capacity).*

**Example 10** *Lifting Molecular Simplicial Complexes to Cell Complexes. Consider $T_1$ as a simplicial complex derived from a molecular structure, where nodes represent atoms,*

*edges represent bonds, and 2-simplices represent stable chemical rings. Suppose we lift this structure to a cell complex $T_2$ that includes larger functional groups such as benzene rings or protein substructures.*

- *The structural lifting $\psi_X$ embeds lower-dimensional simplices into a coarser representation of molecular geometry, grouping functionally related simplices into higher-dimensional cells.*

- *The feature lifting $\psi_F$ ensures that atomic properties (e.g., electronegativity, charge distribution) are mapped to molecular functional groups, enabling efficient coarse-grained learning in topological graph neural networks.*

Lifting maps can be either *fixed* (Bodnar et al., 2021a; Hajij et al., 2023b) or *learnable* (Battiloro et al., 2024a; Bernárdez et al., 2023; Telyatnikov and Scardapane, 2023; Ramamurthy et al., 2023; Kazi et al., 2022), and they may compute or learn both the features on higher-order cells and the structure of the domain itself.

### A.3 Topological Neural Networks

A.3.1 GENERAL DEFINITION

Topological neural networks (TNNs) are neural architectures that process data defined on topological domains. The higher-order message passing paradigm of Hajij et al. (2023b) provides a unifying framework for TNNs, and all networks used in `TopoBench` can be viewed as special cases of this approach.

**Definition 10 ($k$-cochain spaces)** *Let $\mathcal{C}^k(\mathcal{X}, \mathbb{R}^d)$ be the $\mathbb{R}$-vector space of functions $\mathbf{H}_k$ where $\mathbf{H}_k \colon \mathcal{X}^k \to \mathbb{R}^d$ for a rank $k \in \mathbb{Z}_{\geq 0}$. This space is called the $k$-**cochain space**, and elements $\mathbf{H}_k$ in $\mathcal{C}^k(\mathcal{X}, \mathbb{R}^d)$ are the $k$-**cochains** (or $k$-**signals**).*

A $k$-cochain is thus a feature vector associated with each $k$-cell. For a graph, 0-cochains correspond to node features, and 1-cochains correspond to edge features.

**Definition 11 (TNN)** *Let $\mathcal{X}$ be a topological domain. Suppose $\mathcal{C}^{i_1} \times \cdots \times \mathcal{C}^{i_m}$ and $\mathcal{C}^{j_1} \times \cdots \times \mathcal{C}^{j_n}$ are Cartesian products of cochain spaces on $\mathcal{X}$. A **topological neural network (TNN)** is a function*

$$TNN \colon \mathcal{C}^{i_1} \times \cdots \times \mathcal{C}^{i_m} \longrightarrow \mathcal{C}^{j_1} \times \cdots \times \mathcal{C}^{j_n}.$$

A TNN takes as input a collection of cochains $(\mathbf{H}_{i_1}, \ldots, \mathbf{H}_{i_m})$ and produces a collection $(\mathbf{K}_{j_1}, \ldots, \mathbf{K}_{j_n})$. To enable data exchange within a topological domain, one relies on *cochain maps* (e.g., incidence or adjacency matrices) and *neighborhood functions*, described next.

Cochain maps are fundamental operators for data manipulation in topological domains. For $r < k$, incidence matrices $B_{r,k}$ and adjacency matrices $A_{r,k}$ define mappings:

$$B_{r,k} \colon \mathcal{C}^k(\mathcal{X}) \to \mathcal{C}^r(\mathcal{X}), \quad A_{r,k} \colon \mathcal{C}^r(\mathcal{X}) \to \mathcal{C}^r(\mathcal{X}).$$

They collectively redistribute signals across different dimensional cells.

**Definition 12 (Neighborhood function)** *Let $S$ be a nonempty set. A **neighborhood function** on $S$ is a function $\mathcal{N} : S \to \mathcal{P}(\mathcal{P}(S))$ that assigns to each point $x$ in $S$ a nonempty collection $\mathcal{N}(x)$ of subsets of $S$. The elements of $\mathcal{N}(x)$ are called **neighborhoods** of $x$ with respect to $\mathcal{N}$.*

Here, $\mathcal{P}$ denotes the power set operator, where $\mathcal{P}(S)$ is the set of all subsets of $S$. Thus, $\mathcal{P}(\mathcal{P}(S))$ represents the set of all collections of subsets of $S$. This formulation enables the assignment of multiple, potentially overlapping neighborhoods to each point, providing the necessary flexibility to describe diverse neighborhood structures across various topological domains.

The flexibility of neighborhood functions is crucial for representing complex relationships in higher-order topological structures, where elements may exhibit multifaceted connections or interactions. By generalizing the concept of node neighborhoods from graphs to higher-order structures, these functions define how information propagates between different elements in the topological domain. This generalization forms the foundation for extending traditional graph-based algorithms to more complex topological spaces, enabling the modeling of sophisticated relational data.

### A.4 Traditional Message Passing on Graphs

GNNs have emerged as a powerful class of models for processing graph-structured data. While numerous variations of GNN architectures exist Veličković et al. (2018), at their core lies an iterative message passing algorithm that propagates information between the nodes of the graph. This process can be understood in terms of the basic concepts we introduced earlier.

Formally, a graph is defined as a tuple of nodes and edges, $\mathcal{G} = (\mathcal{V}, \mathcal{E})$. In the context of k-cochain spaces introduced in Section A.3.1, we can view node features as 0-cochains and edge features as 1-cochains. We denote by $h_k^t \in \mathbb{R}^d$ the hidden state of a node $k$ at MP iteration $t$, which can be seen as an element of a 0-cochain space.

The neighborhood function for a graph, as per Definition 12, is typically defined as $N(k) = \{v \in \mathcal{V} \mid (k, v) \in \mathcal{E}\}$, representing the one-hop neighborhood of each node.

The MP process consists of three main steps:

1. **Message Generation**: Each node $k$ receives messages from all the nodes in its neighborhood $N(k)$. Messages are generated by applying a message function $m(\cdot)$ to the hidden states of node pairs in the graph.

2. **Message Aggregation**: The generated messages are combined using a permutation invariant aggregation function $\oplus$, as shown in Equation 1. This aggregation can be seen as an application of the neighborhood function concept.

3. **Node Update**: An update function $u(\cdot)$ is used to compute a new hidden state for every node, as shown in Equation 2.

These steps are formalized in the following equations:

$$M_k^{t+1} = \bigoplus_{i \in N(k)} m(h_k^t, h_i^t), \tag{1}$$

$$h_k^{t+1} = u(h_k^t, M_k^{t+1}), \tag{2}$$

where $m(\cdot)$ and $u(\cdot)$ are differentiable functions and consequently may be implemented as neural networks.

This process can be seen as a specific instance of the more general higher-order message passing framework that is introduced in the next section, applied to the case of graphs where we only have 0-cells (nodes) and 1-cells (edges).

## A.5 Higher-Order Message Passing

Higher-Order Message Passing (HOMP) generalizes information propagation techniques to complex topological domains such as hypergraphs, simplicial complexes, and cell complexes. This section introduces a formal framework for HOMP, building on the foundational concepts of k-cochain spaces and neighborhood functions defined earlier. By leveraging the rich relationships captured in these advanced topological representations, this unified approach enables modeling and analyzing intricate multi-way interactions across various topological structures, including both traditional graphs and more complex higher-order domains.

**Extending Message Passing to Higher-Order Domains**   The extension of graph message passing to higher-order domains involves generalizing the concepts of message passing to more complex topological structures. This generalization allows us to capture and process richer relational information that goes beyond pairwise interactions.

In higher-order domains, the notion of a "neighborhood" becomes more complex. Instead of just considering adjacent nodes, we now need to consider relationships between higher-dimensional cells (e.g., edges, faces, volumes). The neighborhood functions we defined earlier play a crucial role in formalizing these complex relationships.

**Higher-Order Message Passing Framework**   With k-cochain spaces providing a way to represent data and neighborhood functions defining relationships, we can now formally define the HOMP procedure. Let $\mathcal{X}$ be a topological domain, and let $\mathcal{N} = \{\mathcal{N}_1, \ldots, \mathcal{N}_n\}$ be a set of neighborhood functions defined on $\mathcal{X}$. Consider a cell $x$ and another cell $y \in \mathcal{N}_k(x)$ for some $\mathcal{N}_k \in \mathcal{N}$. A message $m_{x,y}$ between cells $x$ and $y$ is a computation depending on these two cells or on the data they support. Let $\mathcal{N}(x)$ denote the multi-set $\{\!\{\mathcal{N}_1(x), \ldots, \mathcal{N}_n(x)\}\!\}$, and let $h_x^{(l)}$ represent the data supported on the cell $x$ at layer $l$. HOMP is defined as follows:

$$m_{x,y} = \alpha_{\mathcal{N}_k}(h_x^{(l)}, h_y^{(l)}), \tag{3}$$

$$m_x^k = \bigoplus_{y \in \mathcal{N}_k(x)} m_{x,y}, \quad 1 \le k \le n, \tag{4}$$

$$m_x = \bigotimes_{\mathcal{N}_k \in \mathcal{N}(x)} m_x^k, \tag{5}$$

$$h_x^{(l+1)} = \beta(h_x^{(l)}, m_x). \tag{6}$$

where $\bigoplus$ is a permutation-invariant aggregation function, which is referred to as intra-neighborhood aggregation of $x$, and $\bigotimes$, is an aggregation function called the inter-neighborhood aggregation of $x$. The functions $\alpha_{\mathcal{N}_k}$ and $\beta$ are differentiable functions.

To summarize the HOMP process:

- **Message Generation**: $m_{x,y}$ is the message computed from $x$ to $y$ using the function $\alpha_{\mathcal{N}_k}$.

- **Message Aggregation (intra)**: $m_x^k$ aggregates all messages from the neighbors $y$ in the neighborhood $\mathcal{N}_k(x)$ using the intra-neighborhood function $\bigoplus$.

- **Message Aggregation (inter)**: $m_x$ further aggregates these results across all neighborhoods $\mathcal{N}_k \in \mathcal{N}(x)$ using the inter-neighborhood function $\bigotimes$.

- **Cell Update**: $h_x^{(l+1)}$ updates the data on cell $x$ by combining its current data $h_x^{(l)}$ with the aggregated message $m_x$ using the function $\beta$.

This framework allows for rich information exchange across different dimensions and types of relationships in the topological domain, enabling the modeling of complex, multi-way interactions in various real-world systems.

## Appendix B. Implemented Liftings

This appendix provides a detailed overview of the structural liftings currently implemented within `TopoBench`. Table 3 presents each implemented lifting as a row, specifying the source and destination topological domains involved. Additionally, each row indicates whether the lifting is feature-based or connectivity-based. For an intuitive understanding of these lifting types, please refer to the informal definitions in Section 4.3. For a rigorous mathematical treatment of lifting definitions and their taxonomy, please consult Appendix A.2.

Table 3: List of implemented liftings, each one linked with its description. The domains are: PC for point clouds, G for graphs, HG for hypergraphs, SC for simplicial complexes, CC for cellular complexes, and CCC for combinatorial complexes.

| Lifting name | Source | Dest. | Feat.-based | Conn.-based |
|---|---|---|---|---|
| Clique | G | SC | | ✓ |
| Neighborhood | G | SC | | ✓ |
| Vietoris-Rips | G | SC | ✓ | |
| Graph induced | G | SC | | ✓ |
| Line | G | SC | | ✓ |
| Eccentricity | G | SC | | ✓ |
| DnD | G | SC | ✓ | |
| Random latent clique | G | SC | | ✓ |
| Neighborhood complex | G | SC | | ✓ |
| Alpha complex | PC | SC | ✓ | |
| Random flag complex | PC | SC | | ✓ |
| Cycle | G | CC | | ✓ |
| Discrete configuration | G | CC | | ✓ |
| K-hop | G | HG | | ✓ |
| Expander hypergraph | G | HG | | ✓ |

| Lifting name | Source | Dest. | Feat.-based | Conn.-based |
|---|---|---|---|---|
| KNN | G | HG | | ✓ |
| Kernel | G | HG | ✓ | ✓ |
| Mapper | G | HG | | ✓ |
| Modularity maximization | G | HG | ✓ | ✓ |
| Forman-Ricci curvature | G | HG | | ✓ |
| Voronoi | PC | HG | ✓ | |
| PointNet++ | PC | HG | ✓ | |
| Mixture of Gaussians + MST | PC | HG | ✓ | |
| Simplicial paths | G | CCC | | ✓ |
| Coface | SC | CCC | | ✓ |
| Universal strict | HG | CCC | | ✓ |

Lastly, we refer to the TopoBench Wiki to get a full list of compatible structural liftings from the ICML 2024 TDL Challenge (Bernárdez et al., 2024).

## Appendix C. Further Experimental Details

This appendix provides details on the hyperparameter search methodology, optimization strategy, computational resources used for the experiments, and additional results and analyses.

### C.1 Experiment Configuration and Model Execution

To automate the configuration of `TopoBench` modules, the `hydra` package (Yadan, 2019) is employed. In particular, hierarchical configuration groups and registers facilitate easy use of the library: there is no need to meticulously select each module for any given domain. Simply choosing a dataset and a model automatically configures a full default pipeline, eliminating the need for manual intervention. Model execution and training are further automated by the `lightning` library (Lisa and Bot, 2017), which orchestrates training, validation, and testing while also handling logging and callbacks.

### C.2 Hyperparameter Search

Five splits are generated for each dataset to ensure a fair evaluation across domains, allocating 50% of data for training, 25% for validation, and 25% for testing. An exception is made for the ZINC dataset, which uses predefined splits (Irwin et al., 2012).

Each model (in each domain) has numerous specific hyperparameters that can be tuned to enhance performance. TNNs, in particular, come with additional parameters that could further boost results. To avoid the combinatorial explosion of all possible hyperparameter sets, the search space is restricted to hyperparameters common across every model. A grid-search strategy is used to identify the optimal parameters for each model-dataset combination. Specifically, the encoder hidden dimension is varied over {32, 64, 128}, the encoder dropout over {0.25, 0.5}, the number of backbone layers over {1, 2, 3, 4}, the learning rate over {0.01,

0.001}, and the batch size over {128, 256}. For models in the cellular and simplicial domains, the readout type is also varied between direct readout (DR) and signal down-propagation (SDP). If a model exceeds available GPU memory, the batch size, encoder hidden dimension, and number of backbone layers are reduced until training becomes feasible.

For node-level tasks, validation is conducted after each training epoch, continuing until either the maximum number of epochs is reached or the optimization metric fails to improve for 50 consecutive validation epochs; a minimum of 50 epochs is always enforced. For graph-level tasks, validation is performed every 5 epochs, halting early if validation performance fails to improve for 10 consecutive checks. The optimization uses `torch.optim.Adam` combined with `torch.optim.lr_scheduler.StepLR`, where the step size is 50 and $\gamma = 0.5$. Over 100,000 runs have been executed to obtain the final results. In general, the best hyperparameter set is selected based on the highest average performance across the five validation splits. For ZINC, five different initialization seeds are used to obtain an average performance.

All experiments are conducted on a Linux machine equipped with 256 CPU cores, 1TB of system memory, and 8 NVIDIA A30 GPUs, each with 24GB of GPU memory.

## C.3 Descriptive Summaries of Datasets

Table 4 provides descriptive statistics for each dataset used in the numerical experiments (see Section 5 for details) after lifting to three topological domains: simplicial complexes, cellular complexes, and hypergraphs. The columns labeled 0-cell, 1-cell, 2-cell, and 3-cell show the count of each $n$-cell in the resulting dataset. Specifically, a clique complex lifting is applied to obtain a simplicial domain with a maximum dimension of three, cycle-based lifting is used to obtain a cellular domain with a maximum dimension of two, and $k$-hop lifting (with $k = 1$) is used to lift each graph into a hypergraph.

Table 5 reports additional descriptive statistics for the graph datasets used in the experiments. Specifically, the table includes the dimensionality of the number of classes (set to 1 for regression tasks and to the actual class count for classification tasks), and the number of graphs in each dataset and the initial 0-cell (node) features. Note that, for the illustrative experiments in Section 5, a projected sum feature lifting is applied. Consequently, higher-order cells (e.g., 1-cells, 2-cells, etc.) inherit the same initial feature dimensionality as the 0-cells.

## C.4 Additional Results and Analysis

Table 6 additionally presents results for the CCXN and SCCN networks, which on average perform slightly worse than other models. As shown in Table C.4, the CCXN network performs better when using SDP readout, though not as dramatically as CWN under the same strategy. The SCCN model benefits more from SDP readout than other simplicial-domain models (SCN and SCCNN), showing improvements in 9 out of 21 cases, whereas SCCNN and SCN show improvements in 3 and 5 cases, respectively. Overall, cellular models demonstrate improved performance on 15, 19, and 8 datasets for CCXN, CWN, and CCCN, respectively, when using SDP. In contrast, simplicial models achieve 9, 3, and 5 improvements for SCCN, SCCNN, and SCN, respectively, with the same readout.

Note that for demonstration purposes, only one fixed lifting is applied to transform graphs into each of the considered topological domains, leaving a domain-specific optimal lifting

Table 4: Descriptive summaries of the datasets used in the experiments.

| Dataset | Domain | 0-cell | 1-cell | 2-cell | 3-cell | Num. Hyperedges |
|---|---|---|---|---|---|---|
| | Cellular | 2708 | 5278 | 2648 | 0 | 0 |
| Cora | Simplicial | 2708 | 5278 | 1630 | 220 | 0 |
| | Hypergraph | 2708 | 0 | 0 | 0 | 2708 |
| | Cellular | 3327 | 4552 | 1663 | 0 | 0 |
| Citeseer | Simplicial | 3327 | 4552 | 1167 | 255 | 0 |
| | Hypergraph | 3327 | 0 | 0 | 0 | 3327 |
| | Cellular | 19717 | 44324 | 23605 | 0 | 0 |
| PubMed | Simplicial | 19717 | 44324 | 12520 | 3275 | 0 |
| | Hypergraph | 19717 | 0 | 0 | 0 | 19717 |
| | Cellular | 3371 | 3721 | 538 | 0 | 0 |
| MUTAG | Simplicial | 3371 | 3721 | 0 | 0 | 0 |
| | Hypergraph | 3371 | 0 | 0 | 0 | 3371 |
| | Cellular | 122747 | 132753 | 14885 | 0 | 0 |
| NCI1 | Simplicial | 122747 | 132753 | 186 | 0 | 0 |
| | Hypergraph | 122747 | 0 | 0 | 0 | 122747 |
| | Cellular | 122494 | 132604 | 15042 | 0 | 0 |
| NCI109 | Simplicial | 122494 | 132604 | 183 | 0 | 0 |
| | Hypergraph | 122494 | 0 | 0 | 0 | 122494 |
| | Cellular | 43471 | 81044 | 38773 | 0 | 0 |
| PROTEINS | Simplicial | 43471 | 81044 | 30501 | 3502 | 0 |
| | Hypergraph | 43471 | 0 | 0 | 0 | 43471 |
| | Cellular | 859254 | 995508 | 141218 | 0 | 0 |
| REDDIT-BINARY | Simplicial | 859254 | 995508 | 49670 | 1303 | 0 |
| | Hypergraph | 859254 | 0 | 0 | 0 | 859254 |
| | Cellular | 19773 | 96531 | 77758 | 0 | 0 |
| IMDB-BINARY | Simplicial | 19773 | 96531 | 391991 | 1694513 | 0 |
| | Hypergraph | 19773 | 0 | 0 | 0 | 19773 |
| | Cellular | 19502 | 98903 | 80901 | 0 | 0 |
| IMDB-MULTI | Simplicial | 19502 | 98903 | 458850 | 2343676 | 0 |
| | Hypergraph | 19502 | 0 | 0 | 0 | 19502 |
| | Cellular | 277864 | 298985 | 33121 | 0 | 0 |
| ZINC | Simplicial | 277864 | 298985 | 769 | 0 | 0 |
| | Hypergraph | 277864 | 0 | 0 | 0 | 277864 |
| | Cellular | 24492 | 93050 | 68553 | 0 | 0 |
| Amazon Ratings | Simplicial | 24492 | 93050 | 110765 | 64195 | 0 |
| | Hypergraph | 24492 | 0 | 0 | 0 | 24492 |
| | Cellular | 10000 | 39402 | 28955 | 0 | 0 |
| Minesweeper | Simplicial | 10000 | 39402 | 39204 | 9801 | 0 |
| | Hypergraph | 10000 | 0 | 0 | 0 | 10000 |
| | Cellular | 22662 | 32927 | 10266 | 0 | 0 |
| Roman Empire | Simplicial | 22662 | 32927 | 7168 | 0 | 0 |
| | Hypergraph | 22662 | 0 | 0 | 0 | 22662 |
| | Cellular | OOM | OOM | OOM | OOM | OOM |
| Tolokers | Simplicial | OOM | OOM | OOM | OOM | OOM |
| | Hypergraph | 11758 | 0 | 0 | 0 | 11758 |
| | Cellular | 3224 | 9483 | 6266 | 0 | 0 |
| US-county-demos | Simplicial | 3224 | 9483 | 6490 | 225 | 0 |
| | Hypergraph | 3224 | 0 | 0 | 0 | 3224 |

Table 5: Additional descriptive statistics of the graph datasets used in the experiments.

|  | Dataset | 0-cell dim | Num. classes | Num. graphs |
|---|---|---|---|---|
| | Cora | 1433 | 7 | 1 |
| | Citeseer | 3703 | 6 | 1 |
| | PubMed | 19717 | 500 | 3 |
| | MUTAG | 7 | 2 | 188 |
| | NCI1 | 37 | 2 | 4110 |
| | NCI109 | 38 | 2 | 4127 |
| | PROTEINS | 3 | 2 | 1113 |
| Graph | REDDIT-BINARY | 10 | 2 | 2000 |
| | IMDB-BINARY | 136 | 2 | 1000 |
| | IMDB-MULTI | 89 | 3 | 1500 |
| | ZINC | 21 | 1 | 12000 |
| | Amazon Ratings | 300 | 5 | 1 |
| | Minesweeper | 7 | 2 | 1 |
| | Roman Empire | 300 | 18 | 1 |
| | Tolokers | 10 | 2 | 1 |
| | US-county-demos | 6 | 1 | 1 |

strategy beyond the scope of this paper.[5] Specifically, a clique complex is used for simplicial lifting, cycle-based lifting is used for cellular domains, and $k$-hop lifting (with $k = 1$) is used for hypergraphs. Feature projection is also applied, where the $(n-1)$-cell features are multiplied by the corresponding incidence matrices to generate $n$-cell features.

Finally, Tables 8 and 9 present the number of learnable parameters for each best-performing model configuration and their corresponding runtimes. Overall, these results indicate that TNNs tend to be less efficient in terms of memory usage and computational time compared to their graph-based counterparts. However, there are exceptions: EDGNN and UniGNN2 achieve parameter counts comparable to graph baselines, and among the TNNs, AST and EDGNN stand out as the most efficient on average.

## Appendix D. Higher-Order Datasets

### D.1 Descriptive Summaries of Higher-Order Datasets

Tables 11 and 12 provide descriptive summaries of the higher-order datasets included in `TopoBench`, which spans 13 datasets drawn from a broad range of hypergraph and simplicial benchmark sources.

**Hypergraph datasets:** For the co-authorship networks (Cora-CA and DBLP-CA) and co-citation networks (Cora, Citeseer, and Pubmed), we use the versions provided by Yadati et al. (2019). For 3D object classification, we include the Princeton ModelNet40 (Wu et al., 2015) and National Taiwan University (Chen et al., 2003) datasets, with hypergraphs constructed following the protocols in Feng et al. (2019) and Yang et al. (2020), using both MVCNN (Su et al., 2015) and GVCNN (Feng et al., 2019) features. Additionally, we evaluate

---

5. Learnable liftings may further optimize the predictive capacity of higher-order networks.

Table 6: Cross-domain comparison: results are shown as mean and standard deviation. The best result is bold and shaded in grey, while those within one standard deviation are in blue-shaded boxes.

| | Dataset | GCN | GIN | GAT | AST | EDGNN | UniGNN2 | CCXN | CWN | CCCN | SCCN | SCCNN | SCN |
|---|---|---|---|---|---|---|---|---|---|---|---|---|---|
| Node-level tasks | Cora | 87.09 ± 0.2 | 87.21 ± 1.89 | 86.71 ± 0.95 | **88.92 ± 0.44** | 87.06 ± 1.09 | 86.97 ± 0.88 | 86.79 ± 1.81 | 86.32 ± 1.38 | 87.44 ± 1.28 | 80.86 ± 2.16 | 82.19 ± 1.07 | 82.27 ± 1.34 |
| | Citeseer | 75.53 ± 1.27 | 73.73 ± 1.23 | 74.41 ± 1.75 | 73.85 ± 2.21 | 74.93 ± 1.39 | 74.72 ± 1.08 | 74.67 ± 2.24 | 75.2 ± 1.82 | **75.63 ± 1.58** | 69.6 ± 1.83 | 70.23 ± 2.69 | 71.24 ± 1.68 |
| | Pubmed | 89.4 ± 0.3 | 89.29 ± 0.41 | 89.44 ± 0.24 | **89.62 ± 0.25** | 89.04 ± 0.51 | 89.34 ± 0.45 | 88.91 ± 0.47 | 88.64 ± 0.36 | 88.52 ± 0.44 | 88.37 ± 0.48 | 88.18 ± 0.32 | 88.72 ± 0.5 |
| | Amazon | 49.56 ± 0.55 | 49.16 ± 1.02 | 50.17 ± 0.59 | 50.5 ± 0.27 | 48.18 ± 0.09 | 49.06 ± 0.08 | 48.93 ± 0.14 | **51.9 ± 0.15** | 50.26 ± 0.17 | OOM | OOM | OOM |
| | Empire | 78.16 ± 0.32 | 79.56 ± 0.2 | 84.02 ± 0.51 | 79.5 ± 0.13 | 81.01 ± 0.24 | 77.06 ± 0.2 | 81.44 ± 0.31 | 81.81 ± 0.62 | 82.14 ± 0.0 | 88.27 ± 0.14 | **89.15 ± 0.32** | 88.79 ± 0.46 |
| | Minesweeper | 87.52 ± 0.42 | 87.82 ± 0.34 | 89.64 ± 0.43 | 81.14 ± 0.05 | 84.52 ± 0.05 | 78.02 ± 0.0 | 88.88 ± 0.36 | 88.62 ± 0.04 | 89.42 ± 0.0 | 89.07 ± 0.25 | 89.0 ± 0.0 | **90.32 ± 0.11** |
| | Tolokers | 83.02 ± 0.71 | 80.72 ± 1.19 | **84.43 ± 1.0** | 83.26 ± 0.1 | 77.53 ± 0.01 | 77.35 ± 0.03 | OOM | OOM | OOM | OOM | OOM | OOM |
| | Election | 0.31 ± 0.02 | **0.28 ± 0.02** | 0.29 ± 0.02 | 0.29 ± 0.01 | 0.34 ± 0.02 | 0.37 ± 0.02 | 0.35 ± 0.02 | 0.34 ± 0.02 | 0.31 ± 0.02 | 0.53 ± 0.03 | 0.51 ± 0.03 | 0.46 ± 0.04 |
| | Bachelor | 0.29 ± 0.02 | 0.31 ± 0.03 | **0.28 ± 0.02** | 0.3 ± 0.03 | 0.29 ± 0.02 | 0.31 ± 0.02 | 0.32 ± 0.03 | 0.33 ± 0.03 | 0.31 ± 0.02 | 0.36 ± 0.02 | 0.34 ± 0.03 | 0.32 ± 0.02 |
| | Birth | 0.72 ± 0.09 | 0.72 ± 0.09 | 0.71 ± 0.09 | 0.71 ± 0.08 | **0.7 ± 0.07** | 0.73 ± 0.1 | 0.74 ± 0.11 | 0.72 ± 0.09 | 0.71 ± 0.09 | 0.82 ± 0.09 | 0.79 ± 0.12 | 0.71 ± 0.08 |
| | Death | 0.51 ± 0.04 | 0.52 ± 0.04 | 0.51 ± 0.04 | **0.49 ± 0.05** | 0.52 ± 0.05 | 0.51 ± 0.05 | 0.54 ± 0.06 | 0.54 ± 0.06 | 0.54 ± 0.06 | 0.58 ± 0.06 | 0.55 ± 0.05 | 0.52 ± 0.05 |
| | Income | 0.22 ± 0.03 | 0.21 ± 0.02 | **0.2 ± 0.02** | 0.21 ± 0.02 | 0.23 ± 0.02 | 0.23 ± 0.02 | 0.25 ± 0.03 | 0.25 ± 0.03 | 0.23 ± 0.02 | 0.29 ± 0.03 | 0.28 ± 0.03 | 0.25 ± 0.02 |
| | Migration | 0.8 ± 0.12 | 0.8 ± 0.1 | **0.77 ± 0.13** | 0.78 ± 0.12 | 0.8 ± 0.12 | 0.79 ± 0.12 | 0.85 ± 0.18 | 0.84 ± 0.13 | 0.84 ± 0.12 | 0.91 ± 0.18 | 0.9 ± 0.14 | 0.92 ± 0.2 |
| | Unempl | 0.25 ± 0.03 | **0.22 ± 0.02** | 0.23 ± 0.03 | 0.22 ± 0.02 | 0.26 ± 0.03 | 0.28 ± 0.02 | 0.27 ± 0.03 | 0.25 ± 0.03 | 0.24 ± 0.03 | 0.43 ± 0.04 | 0.43 ± 0.04 | 0.38 ± 0.04 |
| Graph-level tasks | MUTAG | 69.79 ± 6.8 | 79.57 ± 6.13 | 72.77 ± 2.77 | 71.06 ± 6.49 | 80.0 ± 4.9 | 80.43 ± 4.09 | 74.89 ± 5.51 | **80.43 ± 1.78** | 77.02 ± 9.32 | 70.64 ± 5.9 | 76.17 ± 6.63 | 73.62 ± 6.13 |
| | PROTEINS | 75.7 ± 2.14 | 75.2 ± 3.3 | 76.34 ± 1.66 | **76.63 ± 1.74** | 73.91 ± 4.39 | 75.2 ± 1.25 | 75.63 ± 2.57 | 76.13 ± 2.7 | 73.33 ± 2.3 | 75.05 ± 2.76 | 74.19 ± 2.86 | 75.27 ± 2.14 |
| | NCI1 | 72.86 ± 0.69 | 74.26 ± 0.96 | 75.0 ± 0.99 | 75.18 ± 1.24 | 73.97 ± 0.82 | 73.02 ± 0.92 | 74.86 ± 0.82 | 73.93 ± 1.87 | **76.67 ± 1.48** | 76.17 ± 1.39 | 76.6 ± 1.75 | 74.49 ± 1.03 |
| | NCI109 | 72.2 ± 1.22 | 74.42 ± 0.7 | 73.8 ± 0.73 | 73.75 ± 1.09 | 74.93 ± 2.5 | 70.76 ± 1.11 | 75.66 ± 1.3 | 73.8 ± 2.06 | 75.35 ± 1.5 | 75.49 ± 1.39 | **77.12 ± 1.07** | 75.7 ± 1.04 |
| | IMDB-BIN | **72.0 ± 2.48** | 70.96 ± 1.93 | 69.76 ± 2.65 | 70.32 ± 3.27 | 69.12 ± 2.92 | 71.04 ± 1.31 | 70.08 ± 1.21 | 70.4 ± 2.02 | 69.12 ± 2.82 | 70.88 ± 3.98 | 70.88 ± 2.25 | 70.8 ± 2.38 |
| | IMDB-MUL | 49.97 ± 2.16 | 47.68 ± 4.21 | 50.13 ± 3.87 | **50.51 ± 2.92** | 49.17 ± 4.35 | 49.76 ± 3.55 | 47.63 ± 3.45 | 49.71 ± 2.83 | 47.79 ± 3.45 | 49.71 ± 3.7 | 48.75 ± 3.98 | 49.49 ± 5.08 |
| | REDDIT | 76.24 ± 0.54 | 81.96 ± 1.36 | 75.68 ± 1.0 | 74.84 ± 2.68 | 83.24 ± 1.45 | 75.56 ± 3.19 | 82.84 ± 2.54 | **85.52 ± 1.38** | 85.12 ± 1.29 | 74.44 ± 1.74 | 77.24 ± 1.87 | 71.28 ± 2.06 |
| | ZINC | 0.62 ± 0.01 | 0.57 ± 0.04 | 0.61 ± 0.01 | 0.59 ± 0.02 | 0.51 ± 0.01 | 0.6 ± 0.01 | 0.4 ± 0.04 | **0.34 ± 0.01** | 0.34 ± 0.02 | 0.46 ± 0.08 | 0.36 ± 0.02 | 0.53 ± 0.04 |

Table 7: Ablation study comparing the performance of CCXN, CWN, CCCN, SCCN, SC-CNN, and SCN models on various datasets using two readout strategies, direct readout (DR) and signal down-propagation (SDP). SDP generally enhances CWN performance, whereas the effect of SDP on CCCN, SCCNN, and SCN varies based on their internal signal propagation mechanisms. Means and standard deviations of performance metris are shown. The best results are shown in bold for each model and readout type.

| | Dataset | CCXN DR | CCXN SDP | CWN DR | CWN SDP | CCCN DR | CCCN SDP | SCCN DR | SCCN SDP | SCCNN DR | SCCNN SDP | SCN DR | SCN SDP |
|---|---|---|---|---|---|---|---|---|---|---|---|---|---|
| Node-level tasks | Cora | 86.32 ± 1.22 | **86.79 ± 1.81** | 74.95 ± 0.98 | **86.32 ± 1.38** | 87.44 ± 1.28 | **87.68 ± 1.17** | **80.86 ± 2.16** | 80.06 ± 1.66 | **82.19 ± 1.07** | 80.65 ± 2.39 | **82.27 ± 1.34** | 79.91 ± 1.18 |
| | Citeseer | 72.87 ± 1.13 | **74.67 ± 2.24** | 70.49 ± 2.85 | **75.2 ± 1.82** | 73.91 ± 1.58 | **74.91 ± 1.25** | 69.6 ± 1.83 | **70.23 ± 2.69** | 69.03 ± 2.01 | **71.24 ± 1.68** | 70.4 ± 1.53 | |
| | Pubmed | **88.91 ± 0.47** | 88.38 ± 0.38 | 86.94 ± 0.68 | **88.64 ± 0.36** | 88.52 ± 0.44 | **88.67 ± 0.39** | 88.04 ± 0.51 | **88.37 ± 0.48** | **88.18 ± 0.32** | 87.78 ± 0.58 | **88.72 ± 0.5** | 88.62 ± 0.44 |
| | Amazon | **48.93 ± 0.14** | 48.34 ± 0.12 | 45.58 ± 0.25 | **51.9 ± 0.15** | **50.55 ± 0.15** | 50.26 ± 0.17 | OOM | OOM | OOM | OOM | OOM | OOM |
| | Empire | 80.46 ± 0.23 | **81.44 ± 0.31** | 66.13 ± 0.03 | **81.81 ± 0.62** | 82.14 ± 0.00 | **82.51 ± 0.0** | 88.2 ± 0.22 | **88.27 ± 0.14** | **89.15 ± 0.32** | 88.73 ± 0.12 | 85.89 ± 0.34 | **88.79 ± 0.46** |
| | Minesweeper | 88.88 ± 0.36 | **89.76 ± 0.32** | 48.82 ± 0.0 | **88.62 ± 0.04** | 89.42 ± 0.00 | **89.85 ± 0.00** | 88.85 ± 0.00 | **89.07 ± 0.25** | 87.4 ± 0.0 | **89.0 ± 0.00** | **90.32 ± 0.11** | 90.27 ± 0.36 |
| | Election | 0.39 ± 0.05 | **0.35 ± 0.02** | 0.6 ± 0.04 | **0.34 ± 0.02** | 0.31 ± 0.02 | **0.31 ± 0.01** | **0.53 ± 0.03** | 0.57 ± 0.02 | **0.51 ± 0.03** | 0.56 ± 0.04 | **0.46 ± 0.04** | 0.51 ± 0.03 |
| | Bachelor | 0.33 ± 0.03 | **0.32 ± 0.03** | 0.33 ± 0.03 | 0.33 ± 0.03 | 0.32 ± 0.02 | **0.31 ± 0.02** | 0.36 ± 0.02 | **0.34 ± 0.02** | **0.34 ± 0.03** | 0.34 ± 0.03 | **0.32 ± 0.02** | 0.32 ± 0.03 |
| | Birth | 0.80 ± 0.12 | **0.74 ± 0.11** | 0.81 ± 0.11 | **0.72 ± 0.09** | **0.71 ± 0.09** | 0.72 ± 0.05 | **0.82 ± 0.09** | 0.83 ± 0.10 | **0.79 ± 0.12** | 0.83 ± 0.12 | **0.71 ± 0.08** | 0.8 ± 0.11 |
| | Death | 0.57 ± 0.06 | **0.54 ± 0.06** | 0.55 ± 0.05 | **0.54 ± 0.06** | 0.54 ± 0.06 | 0.54 ± 0.06 | 0.58 ± 0.06 | **0.56 ± 0.04** | **0.55 ± 0.05** | 0.58 ± 0.05 | **0.52 ± 0.05** | 0.56 ± 0.05 |
| | Income | 0.25 ± 0.03 | 0.25 ± 0.03 | 0.36 ± 0.04 | **0.25 ± 0.03** | 0.23 ± 0.02 | 0.23 ± 0.02 | 0.29 ± 0.03 | 0.29 ± 0.03 | **0.28 ± 0.03** | 0.31 ± 0.03 | **0.25 ± 0.02** | 0.27 ± 0.02 |
| | Migration | **0.80 ± 0.11** | 0.85 ± 0.18 | 0.9 ± 0.16 | **0.84 ± 0.13** | **0.84 ± 0.10** | 0.84 ± 0.12 | **0.91 ± 0.18** | 0.93 ± 0.17 | **0.90 ± 0.14** | 0.93 ± 0.17 | **0.92 ± 0.20** | 0.96 ± 0.23 |
| | Unempl | 0.28 ± 0.05 | **0.27 ± 0.03** | 0.46 ± 0.04 | **0.25 ± 0.03** | **0.24 ± 0.03** | 0.25 ± 0.03 | **0.43 ± 0.04** | 0.47 ± 0.04 | **0.43 ± 0.04** | 0.45 ± 0.04 | **0.38 ± 0.04** | 0.41 ± 0.03 |
| Graph-level tasks | MUTAG | 69.79 ± 4.61 | **74.89 ± 5.51** | 69.68 ± 8.58 | **80.43 ± 1.78** | **80.85 ± 5.42** | 77.02 ± 9.32 | 70.64 ± 5.90 | **73.62 ± 4.41** | **76.17 ± 6.63** | 70.64 ± 3.16 | 71.49 ± 2.43 | **73.62 ± 6.13** |
| | PROTEINS | **75.63 ± 2.57** | 74.91 ± 1.85 | **76.13 ± 1.80** | 76.13 ± 2.70 | **73.55 ± 3.43** | 73.33 ± 2.30 | **75.05 ± 2.76** | 74.34 ± 3.17 | 74.19 ± 2.86 | **74.98 ± 1.92** | **75.27 ± 2.14** | 74.77 ± 1.69 |
| | NCI1 | 72.43 ± 1.72 | **74.86 ± 0.82** | 68.52 ± 0.51 | **73.93 ± 1.87** | 76.67 ± 1.48 | **77.65 ± 1.28** | **76.42 ± 0.88** | 76.17 ± 1.39 | **76.6 ± 1.75** | 75.6 ± 2.45 | **75.27 ± 1.57** | 74.49 ± 1.03 |
| | NCI109 | 73.22 ± 0.48 | **75.66 ± 1.30** | 68.19 ± 0.65 | **73.8 ± 2.06** | **75.35 ± 1.50** | 74.83 ± 1.18 | **75.49 ± 1.39** | 75.31 ± 1.36 | **77.12 ± 1.07** | 75.43 ± 1.94 | 74.58 ± 1.29 | **75.7 ± 1.04** |
| | IMDB-BIN | **70.08 ± 1.21** | 68.96 ± 2.03 | **70.4 ± 2.02** | 69.28 ± 2.57 | 69.12 ± 2.82 | **69.44 ± 2.46** | **70.88 ± 3.98** | 69.76 ± 3.16 | **70.88 ± 2.25** | 69.28 ± 5.69 | **70.8 ± 2.38** | 68.64 ± 3.90 |
| | IMDB-MUL | 47.63 ± 3.45 | **48.75 ± 3.56** | 49.71 ± 2.83 | **49.87 ± 2.33** | **49.01 ± 2.63** | 47.79 ± 3.45 | **49.71 ± 3.70** | 47.31 ± 3.12 | **48.75 ± 3.98** | 46.67 ± 3.13 | 48.16 ± 2.89 | **49.49 ± 5.08** |
| | REDDIT | 74.40 ± 1.50 | **82.84 ± 2.54** | 76.20 ± 0.86 | **85.52 ± 1.38** | **85.12 ± 1.29** | 83.32 ± 0.73 | 74.16 ± 1.54 | **74.44 ± 1.74** | 75.56 ± 3.46 | **77.24 ± 1.87** | **71.28 ± 2.06** | 69.68 ± 4.0 |
| | ZINC | 0.63 ± 0.02 | **0.40 ± 0.04** | 0.70 ± 0.0 | **0.34 ± 0.01** | 0.35 ± 0.02 | **0.34 ± 0.02** | 0.55 ± 0.01 | **0.46 ± 0.08** | **0.36 ± 0.01** | 0.36 ± 0.02 | 0.59 ± 0.01 | **0.53 ± 0.04** |

Table 8: Model sizes corresponding to the best set of hyperparameters

| Model | GCN | GAT | GIN | AST | EDGNN | UniGNN2 | CWN | CCCN | CCXN | SCN | SCCN | SCCNN |
|---|---|---|---|---|---|---|---|---|---|---|---|---|
| Cora | 234.63K | 113.61K | 105.03K | 60.26K | 113.29K | 109.06K | 343.11K | 451.85K | 735.37K | 144.62K | 155.88K | 164.17K |
| Citeseer | 525.06K | 558.60K | 122.05K | 132.87K | 258.50K | 541.32K | 1754.50K | 1032.84K | 758.41K | 737.29K | 782.34K | 893.13K |
| Pubmed | 114.69K | 148.23K | 53.38K | 280.83K | 147.59K | 114.56K | 163.72K | 85.76K | 277.51K | 134.40K | 457.99K | 605.06K |
| MUTAG | 67.97K | 22.02K | 38.40K | 80.77K | 5.73K | 84.10K | 334.72K | 284.29K | 73.86K | 20.03K | 398.85K | 27.11K |
| PROTEINS | 13.19K | 10.11K | 13.19K | 14.34K | 5.60K | 21.31K | 101.12K | 34.56K | 86.53K | 10.24K | 397.31K | 26.72K |
| NCI1 | 6.72K | 11.20K | 154.37K | 57.47K | 88.19K | 104.32K | 124.10K | 63.87K | 15.87K | 94.88K | 131.84K | 188.99K |
| NCI109 | 23.75K | 11.23K | 154.50K | 221.57K | 88.32K | 4.61K | 412.29K | 17.67K | 71.36K | 26.08K | 135.75K | 49.54K |
| IMDB-BIN | 21.70K | 21.83K | 9.89K | 114.24K | 9.86K | 100.61K | 68.80K | 218.24K | 19.04K | 202.63K | 563.07K | 285.83K |
| IMDB-MUL | 62.08K | 6.37K | 18.76K | 111.30K | 27.01K | 8.32K | 19.68K | 45.64K | 56.26K | 27.94K | 545.16K | 121.22K |
| REDDIT | 13.63K | 30.66K | 10.08K | 106.18K | 5.83K | 68.10K | 26.66K | 47.94K | 57.54K | 7.84K | 69.31K | 286.59K |
| Amazon | 122.37K | 156.16K | 89.35K | 155.91K | 122.24K | 89.22K | 578.95K | 310.15K | 200.96K | OOM | OOM | OOM |
| Minesweeper | 9.28K | 5.89K | 51.46K | 118.02K | 21.70K | 51.33K | 22.24K | 35.07K | 8.83K | 51.97K | 25.15K | 33.44K |
| Empire | 37.39K | 41.81K | 33.23K | 257.17K | 41.49K | 90.90K | 142.74K | 86.10K | 43.83K | 415.89K | 612.50K | 240.53K |
| Tolokers | 84.87K | 151.94K | 3.75K | 217.99K | 21.89K | 13.57K | 12.07K | OOM | OOM | OOM | OOM | OOM |
| Election | 84.22K | 151.30K | 150.27K | 217.34K | 21.57K | 4.58K | 118.08K | 234.37K | 253.18K | 16.64K | 11.52K | 415.10K |
| Bachelor | 17.47K | 151.30K | 7.81K | 316.93K | 84.10K | 3.55K | 43.78K | 34.88K | 12.86K | 27.14K | 43.52K | 415.10K |
| Birth | 4.64K | 30.34K | 5.70K | 30.21K | 84.10K | 13.25K | 26.24K | 22.40K | 253.18K | 103.42K | 11.52K | 26.98K |
| Death | 21.63K | 10.18K | 7.81K | 316.93K | 84.10K | 51.07K | 26.24K | 15.58K | 12.86K | 27.14K | 11.52K | 415.10K |
| Income | 17.47K | 151.30K | 9.92K | 217.34K | 21.57K | 4.58K | 85.18K | 12.42K | 12.86K | 27.14K | 268.03K | 105.15K |
| Migration | 67.71K | 10.18K | 38.27K | 80.64K | 21.57K | 51.07K | 26.24K | 15.58K | 65.15K | 16.64K | 11.52K | 415.10K |
| Unempl | 84.22K | 151.30K | 117.25K | 105.86K | 21.57K | 5.60K | 101.63K | 234.37K | 286.46K | 27.14K | 11.52K | 105.15K |
| ZINC | 22.59K | 22.85K | 10.40K | 106.82K | 22.53K | 102.14K | 88.06K | 287.74K | 16.48K | 24.42K | 617.86K | 1453.82K |
| Average size | 75K ± 111K | 89K ± 119K | 54K ± 53K | 150K ± 88K | 59K ± 60K | 74K ± 109K | 210K ± 367K | 170K ± 229K | 158K ± 212K | 107K ± 172K | 263K ± 251K | 313K ± 340K |

Table 9: Model runtime in seconds corresponding to the best set of hyperparameters

| Model | GCN | GAT | GIN | AST | EDGNN | UniGNN2 | CWN | CCCN | CCXN | SCN | SCCN | SCCNN |
|---|---|---|---|---|---|---|---|---|---|---|---|---|
| Cora | 19.71 ± 3.18 | 33.51 ± 6.33 | 24.24 ± 7.48 | 40.23 ± 11.82 | 28.17 ± 3.97 | 35.28 ± 5.14 | 48.49 ± 27.58 | 39.36 ± 6.72 | 35.74 ± 2.93 | 51.53 ± 14.0 | 84.04 ± 13.39 | 65.62 ± 23.49 |
| Citeseer | 22.58 ± 2.48 | 21.18 ± 2.24 | 24.65 ± 2.58 | 58.78 ± 2.26 | 39.59 ± 1.93 | 41.58 ± 3.55 | 48.89 ± 4.02 | 48.39 ± 4.57 | 53.88 ± 2.81 | 58.85 ± 18.23 | 65.53 ± 13.34 | 80.6 ± 36.33 |
| Pubmed | 40.89 ± 8.39 | 50.2 ± 13.27 | 59.91 ± 10.56 | 84.67 ± 13.03 | 87.25 ± 13.88 | 160.37 ± 31.88 | 147.59 ± 28.96 | 172.63 ± 33.06 | 212.66 ± 52.26 | 151.0 ± 22.55 | 134.82 ± 34.81 | 171.92 ± 48.25 |
| MUTAG | 3.83 ± 0.89 | 4.16 ± 1.05 | 4.6 ± 0.56 | 9.9 ± 3.24 | 5.81 ± 1.13 | 5.5 ± 0.78 | 10.92 ± 0.96 | 12.04 ± 2.21 | 10.76 ± 1.89 | 8.47 ± 2.43 | 10.71 ± 2.92 | 14.06 ± 2.51 |
| PROTEINS | 8.18 ± 2.47 | 8.18 ± 2.3 | 8.88 ± 2.34 | 15.81 ± 2.89 | 15.15 ± 3.3 | 27.8 ± 7.77 | 53.6 ± 17.7 | 41.63 ± 7.23 | 51.98 ± 8.5 | 42.78 ± 13.41 | 70.06 ± 15.91 | 54.13 ± 11.27 |
| NCI1 | 53.23 ± 19.67 | 57.32 ± 17.49 | 61.2 ± 23.97 | 138.13 ± 46.01 | 110.86 ± 27.75 | 169.17 ± 11.25 | 302.34 ± 63.44 | 372.36 ± 109.47 | 244.72 ± 46.09 | 276.32 ± 63.21 | 332.76 ± 51.89 | 307.2 ± 83.01 |
| NCI109 | 37.4 ± 8.63 | 56.44 ± 9.05 | 50.32 ± 7.98 | 138.66 ± 26.38 | 126.61 ± 45.53 | 61.25 ± 16.58 | 294.79 ± 46.27 | 272.3 ± 20.89 | 225.72 ± 86.57 | 226.23 ± 66.29 | 321.76 ± 55.92 | 353.69 ± 105.89 |
| IMDB-BIN | 8.18 ± 3.33 | 7.48 ± 2.21 | 7.72 ± 1.72 | 21.34 ± 2.96 | 13.06 ± 4.76 | 20.75 ± 6.02 | 61.16 ± 9.33 | 51.88 ± 12.3 | 51.2 ± 18.14 | 374.83 ± 132.68 | 432.23 ± 57.29 | 515.13 ± 112.53 |
| IMDB-MUL | 10.27 ± 3.58 | 9.79 ± 1.85 | 10.42 ± 3.93 | 20.89 ± 5.05 | 13.89 ± 3.33 | 39.99 ± 5.18 | 73.6 ± 20.07 | 72.3 ± 13.76 |  | 776.65 ± 147.42 | 716.44 ± 232.31 | 895.77 ± 399.41 |
| REDDIT | 16.17 ± 1.75 | 26.87 ± 5.02 | 28.75 ± 7.64 | 72.33 ± 18.33 | 73.92 ± 26.58 | 307.4 ± 159.71 | 1230.53 ± 270.03 | 1653.47 ± 641.13 | 1435.07 ± 427.2 | 985.1 ± 68.42 | 1622.17 ± 667.71 | 3670.64 ± 423.25 |
| Amazon | 22.12 ± 4.99 | 25.09 ± 4.41 | 15.25 ± 2.73 | 61.77 ± 8.17 | 52.31 ± 4.88 | 264.24 ± 29.28 | 239.31 ± 45.79 | 149.34 ± 58.59 | 201.99 ± 38.94 | OOM | OOM | OOM |
| Minesweeper | 7.33 ± 1.66 | 10.64 ± 1.1 | 8.32 ± 2.51 | 15.79 ± 2.67 | 20.4 ± 4.38 | 58.72 ± 14.39 | 52.12 ± 13.87 | 98.2 ± 15.21 | 50.27 ± 14.89 | 82.5 ± 19.14 | 28.76 ± 3.74 | 54.1 ± 20.66 |
| Empire | 23.38 ± 2.21 | 27.1 ± 1.89 | 22.36 ± 4.9 | 82.61 ± 15.26 | 63.77 ± 13.16 | 122.23 ± 35.71 | 67.26 ± 10.32 | 84.54 ± 6.34 | 63.27 ± 15.66 | 69.98 ± 14.33 | 94.94 ± 27.26 | 96.45 ± 40.6 |
| Tolokers | 163.93 ± 31.44 | 119.87 ± 42.98 | 77.75 ± 26.74 | 117.83 ± 23.21 | 210.78 ± 30.25 | 412.53 ± 94.18 | OOM | OOM | OOM | OOM | OOM | OOM |
| Election | 2.27 ± 0.29 | 2.66 ± 0.43 | 2.1 ± 0.33 | 3.33 ± 0.34 | 2.21 ± 0.34 | 2.97 ± 0.32 | 4.01 ± 0.39 | 4.15 ± 0.28 | 3.98 ± 0.42 | 5.18 ± 0.48 | 4.23 ± 0.73 | 9.35 ± 4.52 |
| Bachelor | 2.28 ± 0.11 | 2.69 ± 0.39 | 1.79 ± 0.28 | 3.09 ± 0.31 | 2.24 ± 0.38 | 2.3 ± 0.25 | 2.99 ± 0.34 | 3.01 ± 0.32 | 3.45 ± 0.29 | 5.11 ± 0.67 | 5.98 ± 1.8 | 7.84 ± 2.14 |
| Birth | 2.04 ± 0.27 | 2.29 ± 0.33 | 1.93 ± 0.32 | 2.18 ± 0.32 | 2.2 ± 0.39 | 2.33 ± 0.29 | 4.11 ± 0.48 | 3.03 ± 0.3 | 4.04 ± 0.45 | 4.99 ± 0.4 | 10.3 ± 5.81 | 29.24 ± 16.97 |
| Death | 2.21 ± 0.3 | 2.1 ± 0.32 | 1.77 ± 0.32 | 3.13 ± 0.24 | 2.35 ± 0.22 | 2.66 ± 0.46 | 4.54 ± 0.55 | 4.08 ± 0.51 | 3.49 ± 0.32 | 4.62 ± 0.55 | 10.31 ± 5.89 | 8.2 ± 1.82 |
| Income | 2.21 ± 0.18 | 2.66 ± 0.41 | 2.03 ± 0.35 | 3.33 ± 0.35 | 2.6 ± 0.33 | 2.94 ± 0.37 | 3.36 ± 0.28 | 3.71 ± 0.28 | 3.42 ± 0.31 | 5.08 ± 0.63 | 10.51 ± 8.3 | 5.57 ± 0.7 |
| Migration | 1.86 ± 0.3 | 2.17 ± 0.3 | 1.87 ± 0.31 | 3.49 ± 0.24 | 2.19 ± 0.3 | 2.69 ± 0.43 | 4.6 ± 0.35 | 4.61 ± 0.36 | 4.48 ± 0.79 | 5.22 ± 0.47 | 4.15 ± 0.76 | 7.79 ± 2.02 |
| Unempl | 2.26 ± 0.35 | 2.68 ± 0.3 | 2.02 ± 0.31 | 4.73 ± 0.31 | 2.17 ± 0.28 | 4.06 ± 0.08 | 3.97 ± 0.3 | 4.19 ± 0.29 | 5.1 ± 0.8 | 5.37 ± 0.44 | 4.17 ± 0.73 | 5.6 ± 0.67 |
| ZINC | 146.89 ± 27.63 | 171.15 ± 64.67 | 168.43 ± 107.1 | 397.77 ± 153.75 | 357.68 ± 46.95 | 520.58 ± 235.2 | 1390.21 ± 96.86 | 1621.11 ± 141.72 | 1226.67 ± 317.55 | 1209.83 ± 571.93 | 1226.19 ± 1686.57 | 2060.2 ± 408.2 |
| Average runtime | 27.23 ± 43.95 | 29.37 ± 42.38 | 26.65 ± 38.89 | 59.08 ± 88.18 | 56.14 ± 85.81 | 102.00 ± 147.61 | 191.18 ± 384.52 | 224.64 ± 479.47 | 188.77 ± 389.49 | 217.48 ± 355.56 | 259.50 ± 444.10 | 420.65 ± 903.88 |

Table 10: TNNs utilized in the experiments and their references

| Acronym | Neural network name | Reference |
|---|---|---|
| **Graph neural networks** | | |
| GAT | Graph attention network | Veličković et al. (2018) |
| GIN | Graph isomorphism network | Xu et al. (2019) |
| GCN | Semi-Supervised Classification with Graph Convolutional Networks | Kipf and Welling (2016) |
| **Simplicial complexes** | | |
| SAN | Simplicial Attention Neural Networks | Giusti et al. (2022) |
| SCCN | Efficient Representation Learning for Higher-Order Data with Simplicial Complexes | Yang et al. (2022) |
| SCCNN | Convolutional Learning on Simplicial Complexes | Yang and Isufi (2023) |
| SCN | Simplicial Complex Neural Networks | Ebli et al. (2020) |
| **Cellular complexes** | | |
| CAN | Cell Attention Network | Giusti et al. (2023) |
| CCCN | Generalized simplicial attention neural networks [6] | Battiloro et al. (2024b) |
| CXN | Cell Complex Neural Networks | Hajij et al. (2020) |
| CWN | Weisfeiler and Lehman Go Cellular: CW Networks | Bodnar et al. (2021a) |
| **Hypergraphs** | | |
| AllSetTransformer | You are AllSet: A Multiset Function Framework for Hypergraph Neural Networks | Chien et al. (2021) |
| EDGNN | Equivariant Hypergraph Diffusion Neural Operators | Wang et al. (2022) |
| UniGNN | UniGNN: a Unified Framework for Graph and Hypergraph Neural Networks | Huang and Yang (2021) |

performance on three datasets with categorical attributes—20Newsgroups, Mushroom, and ZOO—sourced from the UCI Categorical Machine Learning Repository (Dua et al., 2017). For these, we construct hypergraphs as in Yadati et al. (2019), where a hyperedge is formed by grouping data points sharing the same categorical feature value.

Table 11 summarizes the statistics of the hypergraph datasets along with their associated homophily metrics: clique-expansion homophily (Wang et al., 2023) and $\Delta$-homophily (Telyatnikov et al., 2025). These measures capture the degree to which hyperedges align with label information, serving as indicators of how well the hypergraph structure supports downstream classification.

**Simplicial datasets:** In contrast, the MANTRA family (Ballester et al., 2024) comprises purely topological datasets of 2-manifold triangulations. From the suite of tasks it offers, we focus on three representative classification problems: (1) NAME: predicting the homeomorphism class of a triangulated surface, (2) ORIENT: determining its orientability, and (3) $\beta_1$, $\beta_2$: predicting the values of the first and second Betti numbers. The task of predicting Betti numbers is performed as a regression, while the outputs are rounded, and then classification metrics are employed to assess performance. Table 12 reports the corresponding dataset statistics for this family.

Table 11: Statistics of hypergraph higher-order datasets

| Dataset | 0-cell dim | Num. 0-cell | Num. 1-cell | Num. classes | CE Homophily | $\frac{1}{|\mathcal{V}|}\sum_{v\in\mathcal{V}} h_v^0$ | $\frac{1}{|\mathcal{V}|}\sum_{v\in\mathcal{V}} h_v^1$ |
|---|---|---|---|---|---|---|---|
| Cora | 1433 | 2708 | 1579 | 7 | 89.74 | 84.10 | 78.08 |
| Citeseer | 3703 | 3312 | 1079 | 6 | 89.32 | 78.25 | 74.18 |
| Pubmed | 500 | 19717 | 7963 | 3 | 95.24 | 82.05 | 75.73 |
| CORA-CA | 1433 | 2708 | 1072 | 7 | 80.26 | 80.81 | 76.51 |
| DBLP-CA | 1425 | 41302 | 22363 | 6 | 86.88 | 88.86 | 86.01 |
| ZOO | 16 | 101 | 43 | 7 | 82.88 | 91.13 | 85.79 |
| 20Newsgroups | 100 | 16262 | 100 | 4 | 75.25 | 81.26 | 74.78 |
| Mushroom | 22 | 8124 | 298 | 2 | 85.33 | 88.05 | 84.41 |
| NTU2012 | 100 | 16242 | 2012 | 67 | 46.07 | 53.24 | 41.95 |
| ModelNet40 | 100 | 12311 | 12311 | 40 | 24.07 | 42.16 | 29.42 |

Table 12: Statistics of MANTRA family simplicial datasets

| Dataset | 0/1/2-cell dim | Avg. num. 0-cell | Avg. num. 1-cell | Avg. num. 2-cell | Num. classes | Num. objects |
|---|---|---|---|---|---|---|
| NAME | 1/1/1 | 9.98 | 34.43 | 22.95 | 8 | 43138 |
| ORIENT | 1/1/1 | 9.98 | 34.43 | 22.95 | 2 | 43138 |
| $\beta_1,\beta_2$ | 1/1/1 | 9.98 | 34.43 | 22.95 | 1 | 43138 |

## D.2 Hypergraph Higher-Order Datasets Results

Table 13 demonstrates varying model effectiveness across real-world classification hypergraph datasets. While no single model consistently outperforms others across all datasets, AllSet-Transformer achieves the best performance in 5 out of 10 cases. UniGNN2 achieves top performance on several datasets, including Cora and ModelNet40, while EDGNN leads on CORA-CA and Citeseer. It is important to note that the results shown in Table 13 and Table 1 for Cora, Citeseer, and Pubmed refer to the same base datasets but differ in the nature of their topology. Specifically, the higher-order structures in Table 1 are derived

via lifting mechanisms applied to graph data (graph representation of the Cora, Citeseer, and Pubmed), whereas the results in Table 13 are obtained from real hypergraph datasets, where hyperedges are constructed based on available metadata, please refer to Appendix D.2 of Telyatnikov et al. (2025).

Table 13: Test accuracy (mean ± std) for each hypergraph dataset (rows) and model (columns). The best result is bold and shaded in gray, while those within one standard deviation are in blue-shaded boxes.

|            | EDGNN          | AllSetTransformer | UniGNN2         |
|------------|----------------|-------------------|-----------------|
| Cora       | 78.14 ± 0.72   | 78.91 ± 1.06      | **79.56 ± 1.54** |
| Citeseer   | **72.58 ± 1.51** | 71.57 ± 1.71    | 72.39 ± 2.38    |
| Pubmed     | 87.04 ± 0.34   | **87.22 ± 0.28**  | 86.93 ± 0.53    |
| CORA-CA    | **82.36 ± 0.72** | 82.19 ± 2.61    | 81.71 ± 1.42    |
| DBLP-CA    | 90.83 ± 0.25   | **91.98 ± 0.18**  | 90.72 ± 0.23    |
| Zoo        | 86.92 ± 6.99   | **90.77 ± 8.85**  | 90.77 ± 9.65    |
| 20newsW100 | 79.96 ± 0.77   | **81.04 ± 0.72**  | 80.21 ± 0.75    |
| Mushroom   | 99.78 ± 0.07   | **99.93 ± 0.03**  | 99.61 ± 0.21    |
| NTU2012    | 87.55 ± 1.52   | **89.07 ± 0.90**  | 88.47 ± 1.92    |
| ModelNet40 | 98.27 ± 0.21   | 98.18 ± 0.12      | **98.41 ± 0.11** |

## D.3 Simplicial Higher-Order Datasets

Table 14 reports results introduced by Carrasco et al. (2025), a study conducted within the TopoBench framework, on real-world simplicial higher-order datasets. SCCNN achieves the highest overall performance, reaching 95.08% accuracy on the NAME classification task, while also maintaining strong performance across other metrics. Simplicial complex-based networks (SCN, SCCNN, SaNN, GCCN) consistently outperform standard graph methods (GCN, GAT, GIN), with SCCNN exhibiting particularly stable results, indicated by a low standard deviation of 0.56. These findings highlight the advantage of topological networks in modeling higher-order tasks, where conventional pairwise graph structures fall short in capturing complex relational patterns.

Table 14: Higher-order datasets. Results are shown as mean ± standard deviation. The best result is bold and shaded in grey, while those within one standard deviation are in blue-shaded boxes.

|            | Model   | NAME (↑)        | ORIENT (↑)      | $\beta_1$ (↑)    | $\beta_2$ (↑)     |
|------------|---------|-----------------|-----------------|-----------------|------------------|
| Graph      | GCN     | 42.14 ± 2.72    | 47.94 ± 0.00    | 46.86 ± 4.50    | 0.00 ± 0.00      |
|            | GAT     | 18.09 ± 0.65    | 47.94 ± 0.00    | 7.45 ± 0.05     | 0.00 ± 0.00      |
|            | GIN     | 76.14 ± 0.14    | 56.28 ± 0.45    | 88.13 ± 0.00    | 0.93 ± 1.21      |
| Simplicial | SCN     | 79.48 ± 1.36    | 69.55 ± 0.97    | 76.45 ± 3.06    | 5.45 ± 2.31      |
|            | SCCNN   | **95.08 ± 0.56** | **86.29 ± 1.23** | 90.20 ± 0.20   | 65.82 ± 2.70     |
|            | SaNN    | 81.76 ± 1.37    | 61.65 ± 0.55    | 88.46 ± 0.09    | 39.22 ± 2.80     |
|            | GCCN    | 86.76 ± 1.27    | 76.60 ± 1.67    | 84.20 ± 4.80    | 41.82 ± 20.19    |
|            | HOPSE-M | 91.50 ± 1.45    | 80.68 ± 1.72    | **90.26 ± 0.55** | **71.69 ± 1.50** |
|            | HOPSE-G | 81.75 ± 1.26    | 62.17 ± 0.98    | 88.28 ± 0.08    | 35.37 ± 2.25     |

# Appendix E. Additional Dataset Details

To promote transparency, reproducibility, and ease of use, all dataset loading and preprocessing functionalities are encapsulated within the `TopoBench` library's loader module (loader module). Most graph datasets are processed using the official `torch_geometric` loaders, which parse raw formats (e.g., `.csv`, `.npz`, or edge lists). The US-county-demos dataset is further adapted from the following repository (link). Higher-order hypergraph datasets are sourced from the repository of Chien et al. (2021) (link), while the MANTRA family datasets are adapted from (link) and integrated into `TopoBench` with consistent formatting adapted from `torch_geometric`, which stores the preprocessed datasets in the standardized PyTorch `.pt` files. This unified pipeline automates the full dataset preparation process and removes the need to access external repositories manually.

   `TopoBench` includes all dataset licenses—where applicable—in the file located at the root of the repository and named `third_party_licenses.txt`. Additionally, a dedicated Datasets section in the `README.md` file provides references to the original source papers for each dataset included in the benchmark.

## E.1 Graph datasets

**Shared preprocessing:** As emphasised in Appendix C.4–for demonstration purposes, only one fixed lifting is applied to transform graphs into each of the considered topological domains, leaving a domain-specific optimal lifting strategy beyond the scope of this paper. Specifically, a clique complex is used for simplicial lifting, cycle-based lifting is used for cellular domains, and $k$-hop lifting (with $k = 1$) is used for hypergraphs. Feature projection is also applied, where the $(n - 1)$-cell features are multiplied by the corresponding incidence matrices to generate $n$-cell features.

   Cora, Citeseer, and Pubmed are adapted from the open-source Planetoid dataset collection available in the `torch_geometric` repository (link to dataset). *Preprocessing:* no additional preprocessing is applied beyond the shared one.

   MUTAG, PROTEINS, REDDIT-BINARY, IMDB-BINARY, IMDB-MULTI, NCI1, and NCI109 are sourced from the open-source TUDataset collection (link to dataset). *Preprocessing:* no additional preprocessing is applied beyond the shared one.

   ZINC is adapted from the open-source dataset available at (link to dataset). *Preprocessing:* node features are first transformed into one-hot encodings, after which the shared preprocessing is applied.

   Amazon Ratings, Minesweeper, Roman Empire, and Tolokers are obtained using the `HeterophilousGraphDataset` loader from `torch_geometric` (link to dataset). *Preprocessing:* no additional preprocessing is applied beyond the shared one.

   The US-county-demos dataset is taken from the official Cornell website (link to dataset). *Preprocessing:* the version used is already preprocessed as in Jia and Benson (2020), and no further preprocessing is applied beyond the shared one.

## E.2 Higher-Order datasets

Higher-order hypergraph datasets are acquired from the repository of Chien et al. (2021) (link to dataset) and adapted to the benchmark pipeline to conform to the integrated

`torch_geometric` format used in `TopoBench`. Additional information regarding the hypergraph datasets is provided and discussed in Appendix D.1.

The code for the MANTRA family datasets is adapted from the (https://github.com/aidoslab/MANTRA) and integrated into `TopoBench` following the `torch_geometric` formatting.

