# OpenReview forum: "TopoBench: A Framework for Benchmarking Topological Deep Learning"
_DMLR — Accepted by DMLR_

### Review · Reviewer_gAuT · 2025-05-12

**Recommendation:** 3
**Confidence:** 3

**Summary Of Contributions:**

TopoBench presents a modular, open-source benchmarking framework tailored for topological deep learning (TDL). It decomposes the TDL pipeline into independently configurable components — from data loading and topological lifting to model training and evaluation — enabling reproducibility, flexibility, and cross-domain comparisons. The framework introduces standardized evaluation tools across graph, hypergraph, simplicial, and cell complex domains, supports automated dataset lifting, and offers extensive benchmarking experiments involving 12 topological neural networks (TNNs) across 22 datasets and multiple learning tasks.

**Strengths:**

1. The architecture of TopoBench supports seamless integration and evaluation across a wide range of topological models and datasets.

2. First of its kind to systematically benchmark TNNs across diverse topological domains using unified lifting and preprocessing.


3. Incorporates tools for data preprocessing caching, deterministic splits, and experiment tracking (e.g., Lightning, wandb).


4. Offers tutorials and clear interfaces that lower the barrier to entry for TDL newcomers.

5. Includes an extensive set of experiments and ablation studies (e.g., comparing readout strategies), substantiating claims with robust empirical support.

**Audience:**

Yes

**Broader Impact Concerns:**

No immediate concerns arise regarding ethical risks or societal harm. The authors proactively address ethical standards and contributor conduct. However, broader deployment of TDL in sensitive domains (e.g., social networks, biological data) may require ongoing ethical scrutiny, particularly regarding fairness and explainability in topologically informed predictions.

**Claims And Evidence:**

1. Extensive benchmarking shows performance trends across models and domains.

2.  Ablation studies clarify architectural impact and readout strategies.

3. Reproducibility is demonstrably ensured through public code and tutorials.

**Datasets And Benchmarks:**

Sufficient detail is provided. The datasets are well-documented, accessible, and integrated through a uniform interface. Custom dataset integration is also supported and well-explained.

**Extended Submissions:**

There is no indication that this work is an extension of previously published work. It appears to be original and self-contained.

**Limitations:**

1. Limits the exploration of task-optimized topological representations.

2. Focuses primarily on task-level performance metrics without assessing topological structure learning quality.

3. The effectiveness of fixed lifting strategies may not generalize across tasks and domains.

4. OOM issues suggest scalability limitations for some model-dataset combinations.

**Requested Changes:**

1. Discuss plans or ongoing work to support data-driven lifting mechanisms.

2. Add or at least discuss integrating TDL-specific evaluation criteria like expressivity, spectral efficiency, or topological invariance.


3. Provide explanations or guidelines for dealing with out-of-memory issues, especially for larger datasets or higher-order models.

4. Expand benchmark datasets beyond lifted graphs to include naturally occurring higher-order datasets.

5. Justify the choice of lifting operations used for each domain in the experiments — are they optimal or just illustrative?

**Strengths And Weaknesses:**

1. Current benchmarking results rely on fixed, non-learnable topological liftings. This limits insight into the full potential of TNNs with data-adaptive topological representations.

2. Several models fail to run on certain datasets due to memory constraints, which may limit generality or introduce bias in performance conclusions.

3. The framework mostly uses standard metrics like accuracy and MSE, omitting TDL-specific metrics (e.g., topological expressivity, robustness).

4. While some hypergraph and simplicial datasets are included, many are lifted from graph data; this limits ecological validity.

---

### Review · Reviewer_jrje · 2025-05-25

**Recommendation:** 2
**Confidence:** 2

**Summary Of Contributions:**

This paper presents TopoBench, which is a new open-source library for topological deep learning (TDL). It consists of different modules for data generation, loading, transformation, processing, model training, optimization, and evaluation. This modular design makes it easy to modify and evaluate various TDL pipelines. The goal is to standardize TDL. The authors also conduct experiments using TopoBench to compare graph neural nets and topologic neural nets.

**Strengths:**

S1-S3

**Audience:**

Yes

**Claims And Evidence:**

The contributions claimed in Introduction is somewhat supported, but there are limitations to the supporting evidence (see W1-W4).

**Datasets And Benchmarks:**

The paper's primary contribution appears to be its standardized framework. However, the benchmark section mostly utilizes existing datasets, and critical information regarding these datasets is either incomplete or missing. Specifically, the authors failed to include detailed formats and descriptions (there is a only table in the appendix that lack essential details like the number of features or classes for classification problems), direct web links to data repositories (only linking to the paper itself), comprehensive processing methodologies, licensing information, or instructions on dataset utilization. While details and links to code repositories for example benchmarking are provided to support reproducibility, the lack of thorough dataset documentation significantly hinders the utility and transparency of the benchmark.

**Extended Submissions:**

N/A

**Limitations:**

W1-W4

**Requested Changes:**

W1-W4

**Strengths And Weaknesses:**

S1. Identified an important limitation of existing Topological Deep
Learning studies.

S2. This paper introduces a standardized, modular, and easily extensible framework.

S3. This paper is well written and easy to follow.

W1. The contribution of this paper is unclear. The authors mentioned the lack of datasets and potential to use liftings to create datasets, but did not give concrete examples. In section 3, the authors discussed existing software for TDL, but did not discuss how Topobench is related to those existing software. What are missing from the existing software that the TopoBench is trying to solve? Does TopoBench complement existing works? or how does TopoBench compare with those existing software?

W2. The paper's primary contribution appears to be its standardized framework. However, the benchmark section mostly utilizes existing datasets, and critical information regarding these datasets is either incomplete or missing. Specifically, the authors failed to include detailed formats and descriptions (there is a only table in the appendix that lack essential details like the number of features or classes for classification problems), direct web links to data repositories (only linking to the paper itself), comprehensive processing methodologies, licensing information, or instructions on dataset utilization. While details and links to code repositories for example benchmarking are provided to support reproducibility, the lack of thorough dataset documentation significantly hinders the utility and transparency of the benchmark.

W3. Although this paper is positioned as "a Framework for Benchmarking Topological Deep Learning", the experiments in section 5 are mainly using graph datasets, and did not show the result with datasets with higher-order interactions.

W4. The experiments in Section 5 fail to provide important computational efficiency details, such as running time, memory cost, and the specific experimental environment (CPU, GPU, memory, etc). This information is highly important for evaluating different algorithms.

---

### Review · Reviewer_SaLm · 2025-05-25

**Recommendation:** 4
**Confidence:** 2

**Summary Of Contributions:**

TopoBench addresses key challenges in the rapidly evolving field of TDL, such as the lack of standardized evaluation protocols, the scarcity of higher-order datasets, and the difficulty in comparing different Topological Neural Networks (TNNs). The framework decomposes the TDL pipeline into a sequence of modular components: data generation/loading, transformation/processing (including "lifting" operations), model training, optimization, and evaluation.

**Strengths:**

1. TopoBench provides a much-needed standardized framework for a field (TDL) that currently lacks unified benchmarking practices, thus promoting reproducibility and fair comparisons.
2. The modular design (data, model, training, communication modules) makes it flexible and easier for researchers to integrate new datasets, lifting methods, TNN models, and evaluation metrics.
3. The comprehensive support for various lifting operations is a significant strength. It enables the creation of higher-order datasets from more commonly available graph or point cloud data, democratizing access to TDL.
4. The framework supports a wide range of topological domains (graphs, simplicial complexes, cell complexes, hypergraphs, combinatorial complexes), various TDL models, and diverse datasets.
5.  Being open-source with promised documentation and tutorials lowers the barrier to entry and encourages community adoption and contribution.

**Audience:**

Yes

**Claims And Evidence:**

yes

**Datasets And Benchmarks:**

yes

**Extended Submissions:**

N/A

**Limitations:**

See Weakness.

**Requested Changes:**

1. The paper acknowledges that currently implemented liftings are fixed (pre-defined transformations). Integrating learnable lifting mechanisms, where the transformation to a higher-order domain is itself optimized as part of the learning process, could significantly enhance model performance and representation power.
2. TopoBench facilitates the creation of higher-order data via liftings. However, the inherent scarcity of large-scale, real-world datasets that are natively higher-order remains a challenge for the field. Expanding the built-in collection of such datasets would be beneficial.
3. The current experiments primarily use standard machine learning metrics (accuracy, MSE, MAE). Incorporating evaluation metrics specifically designed to capture the unique advantages of TDL (e.g., expressivity related to topological features, robustness to certain perturbations, explainability derived from topological structures) would provide deeper insights.
4. With numerous lifting methods available, guidance or automated methods for selecting the most appropriate lifting for a given dataset and task would be valuable for users, as this choice can significantly impact performance. The paper uses one fixed lifting per domain for experiments.

**Strengths And Weaknesses:**

Pros:

1. TopoBench provides a much-needed standardized framework for a field (TDL) that currently lacks unified benchmarking practices, thus promoting reproducibility and fair comparisons.
2. The modular design (data, model, training, communication modules) makes it flexible and easier for researchers to integrate new datasets, lifting methods, TNN models, and evaluation metrics.
3. The comprehensive support for various lifting operations is a significant strength. It enables the creation of higher-order datasets from more commonly available graph or point cloud data, democratizing access to TDL.
4. The framework supports a wide range of topological domains (graphs, simplicial complexes, cell complexes, hypergraphs, combinatorial complexes), various TDL models, and diverse datasets.
5.  Being open-source with promised documentation and tutorials lowers the barrier to entry and encourages community adoption and contribution.

Weakness:

1. The paper acknowledges that currently implemented liftings are fixed (pre-defined transformations). Integrating learnable lifting mechanisms, where the transformation to a higher-order domain is itself optimized as part of the learning process, could significantly enhance model performance and representation power.
2. TopoBench facilitates the creation of higher-order data via liftings. However, the inherent scarcity of large-scale, real-world datasets that are natively higher-order remains a challenge for the field. Expanding the built-in collection of such datasets would be beneficial.
3. The current experiments primarily use standard machine learning metrics (accuracy, MSE, MAE). Incorporating evaluation metrics specifically designed to capture the unique advantages of TDL (e.g., expressivity related to topological features, robustness to certain perturbations, explainability derived from topological structures) would provide deeper insights.
4. With numerous lifting methods available, guidance or automated methods for selecting the most appropriate lifting for a given dataset and task would be valuable for users, as this choice can significantly impact performance. The paper uses one fixed lifting per domain for experiments.